# Development of visual cortex in human neonates is selectively modified by postnatal experience

**Mingyang Li[1], Tingting Liu[1], Xinyi Xu[1], Qingqing Wen[1], Zhiyong Zhao[1], Xixi Dang[2], Yi Zhang[1], Dan Wu[1,3,4]\***

[1]Key Laboratory for Biomedical Engineering of Ministry of Education, Department of Biomedical Engineering, College of Biomedical Engineering & Instrument Science, Zhejiang University, Hangzhou, China; [2]Department of Psychology, Zhejiang Sci-Tech University, Hangzhou, China; [3]Children's Hospital School of Medicine, Zhejiang University, Hangzhou, China; [4]Binjiang Institute of Zhejiang University, Hangzhou, China

**Abstract** Experience-dependent cortical plasticity is a pivotal process of human brain development and essential for the formation of most cognitive functions. Although studies found that early visual experience could influence the endogenous development of visual cortex in animals, little is known about such impact on human infants. Using the multimodal MRI data from the developing human connectome project, we characterized the early structural and functional maps in the ventral visual cortex and their development during neonatal period. Particularly, we found that postnatal time selectively modulated the cortical thickness in the ventral visual cortex and the functional circuit between bilateral primary visual cortices. But the cortical myelination and functional connections of the high-order visual cortex developed without significant influence of postnatal time in such an early period. The structure–function analysis further revealed that the postnatal time had a direct influence on the development of homotopic connection in area V1, while gestational time had an indirect effect on it through cortical myelination. These findings were further validated in preterm-born infants who had longer postnatal time but shorter gestational time at birth. In short, these data suggested in human newborns that early postnatal time shaped the structural and functional development of the visual cortex in selective and organized patterns.

**\*For correspondence:**
danwu.bme@zju.edu.cn

## Editor's evaluation

We believe that this study will make significant contributions to developmental neuroscience and vision science as it is a novel attempt to study the processes that might be innate or genetically wired and those that emerge due to worldly experiences within the sensory systems. The authors suggest that early postnatal experience and time spent inside the womb differentially shape the structural and functional development of the visual cortex. The use of large neonatal dataset from the developmental Human Connectome Project is impressive and strengthens the claims made in the paper.

## Introduction

A fundamental question in neuroscience is about the role of experience in neurodevelopment (*Arcaro and Livingstone, 2021*; *Barlow, 1975*; *Holtmaat and Svoboda, 2009*; *Nithianantharajah and Hannan, 2006*). Vision, given its ecological universality and importance across species, has long been

taken as a representative modality to investigate such question (*Barlow, 1975*; *Crair et al., 1998*; *Gödecke and Bonhoeffer, 1996*; *Li et al., 2008*; *Li et al., 2006*; *Roy et al., 2020*), and a framework consisting of two distinct phases was proposed to describe the development of visual cortex (*Barlow, 1975*; *Li et al., 2006*; *White and Fitzpatrick, 2007*). This framework includes an early, experience-independent phase in which the basic layout of neural map is established, and a subsequently experience-dependent phase in which the visual experience refines and shapes the initial neural map. Recent studies further revealed the importance of visual experience during the early period of cortical development, suggesting the intricate interaction between early sensory experience and the endogenous mechanisms in neurodevelopment (*Li et al., 2008*; *Li et al., 2006*; *Roy et al., 2020*). However, these studies were carried out on the model animals (e.g., cat and ferret) using electrophysiological techniques. Due to the lack of noninvasive methods to probe the human infant brain, the influence of early postnatal experience on the development of visual cortex in human infants remains unclear.

Neuroimaging techniques, especially magnetic resonance imaging (MRI), provided an ideal tool to noninvasively measure both the brain structure and function, which, however, remains challenging for the infants' brain due to subject motion, limited resolution, and difficulty in patient recruitment (*Cordero-Grande et al., 2018*). Owing to the recent technical advances in both images acquisition and processing methods (*Hughes et al., 2017*), MRI studies of the perinatal brain, for example, the developing human connectome project (dHCP; http://www.developingconnectome.org/) was initiated to investigate the early structural and functional development of human cerebral cortex. Herein we collected a dataset of multimodal MRI data of 407 neonatal subjects from dHCP to address the above question. The different time intervals between gestational age (GA) at birth and the postmenstrual age (PMA) at scan indicated the difference in the postnatal experience across neonates, which reflects the individual variation of the visual experience. Thus, this time window provides us the opportunity to investigate the contributions of early visual experience versus endogenous neurodevelopment on the development of visual cortex in human newborns.

The main purpose of this study is to describe the early structural and functional development of the ventral visual cortex in human newborns and estimate the contribution of postnatal time (PT) to this process. Particularly, the cortical thickness (CT, *Lyall et al., 2015*; *Sowell et al., 2004*) and T1w/T2w-based cortical myelination (CM, *Glasser and Van Essen, 2011*; *Soun et al., 2017*) data from dHCP was used to measure the development of cortical morphology and microstructure, respectively. The resting-state functional MRI (r-fMRI, *Fitzgibbon et al., 2020*) data was used to evaluate the development of cortical circuits based on functional connectivity between cortical areas. We focused on the primary visual cortex (V1) and higher-level visual cortex, namely, the ventral occipital temporal cortex (VOTC, *Bi et al., 2016*; *Grill-Spector and Weiner, 2014*). The VOTC contains function-specific regions for biologically important categories such as faces (*Kanwisher et al., 1998*), bodies (*Downing et al., 2001*), and scenes (*Epstein and Kanwisher, 1998*), making this region a critical component in the ventral pathway of visual processing (*Bi et al., 2016*; *Grill-Spector and Malach, 2004*; *Grill-Spector and Weiner, 2014*; *Kanwisher, 2010*), and the impact of early visual experience on those regions is not fully understood.

Previous studies have described the developmental trajectory of CT and CM in human infants and found a generally increasing trend for both two measurements with PMA in whole brain (*Bozek et al., 2018*; *Fenchel et al., 2020*). But the contribution of PT to the development was not clarified because PMA reflects both prenatal and postnatal factors. Using r-fMRI, studies found that the primate newborns at a few days of age already had a proto-organization in visual system (*Arcaro and Livingstone, 2017*) and human infants within a few weeks of age showed a category-specific network (*Kamps et al., 2020*). However, the small sample size of those studies might undermine reliability of the results given the well-known instability of r-fMRI signals (*Poldrack et al., 2017*). Moreover, the human infants participating in a previous study (*Kamps et al., 2020*) were around 1 month old (mean age: 27 days; range from 6 to 57 days), who might already acquire some visual experience, and thus this study could not exclude postnatal visual experience on the innate functional connectivity.

In this study, we first characterized the general development of structural morphology (CT) and microstructure (CM) in the ventral cortex and estimate the contributions of prenatal time and PT on the structural development. Then, we used r-fMRI to characterize the innate organization of the ventral cortex directly after birth and investigated the effect of PT on functional networks of V1 and VOTC areas. Furthermore, we carried out a mediation analysis to investigate the relationship between

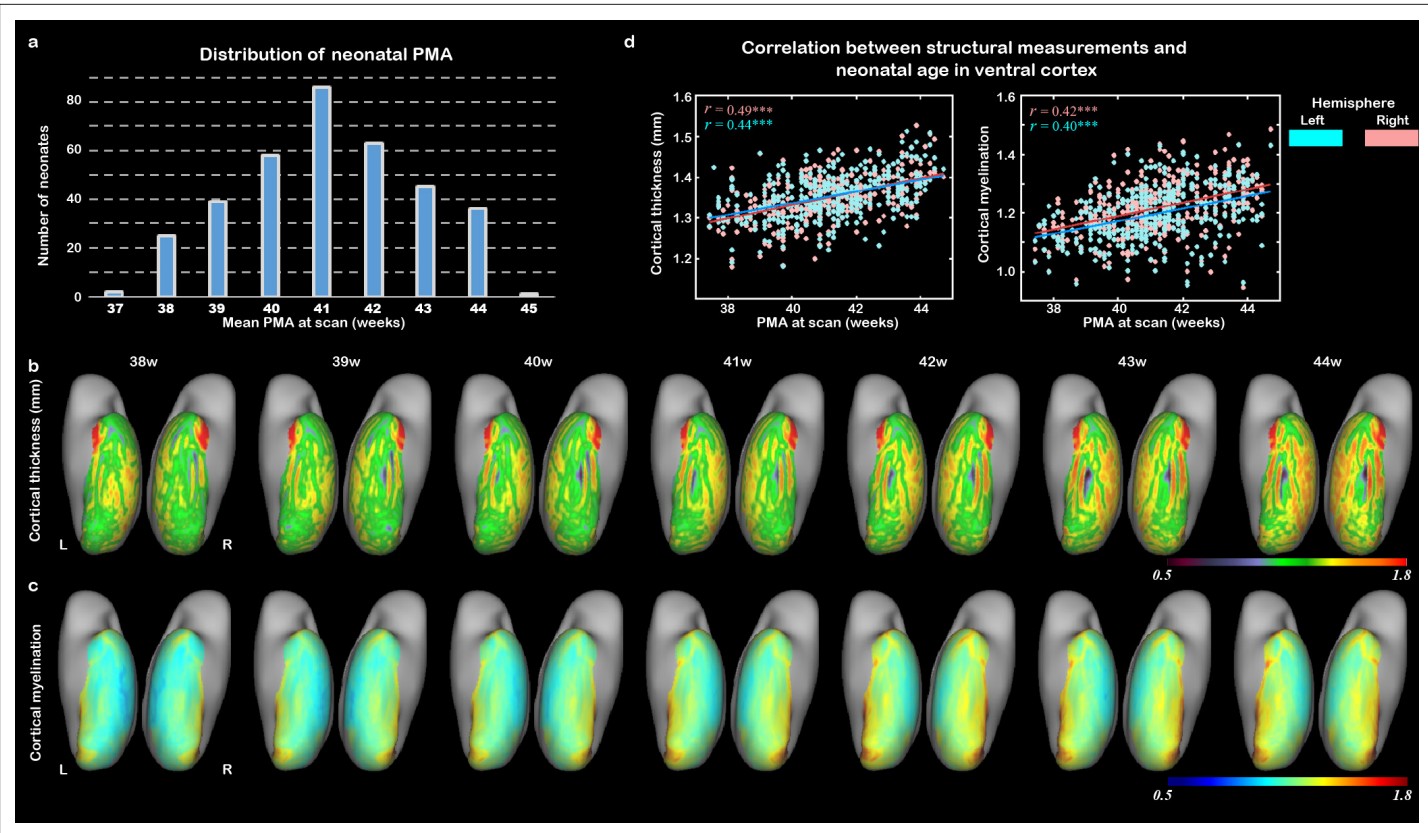

**Figure 1.** The development of cortical structural properties in human newborns. (**a**) The distribution of neonatal postmenstrual age (PMA) at scan in the study population. (**b**) The averaged cortical thickness (CT) and (**c**) the averaged cortical myelination (CM) from 38 to 44 weeks of PMA in the ventral cortex. (**d**) The correlation between CT/CM and PMA in right (peach) and left (blue) ventral cortex (***p<0.001).

The online version of this article includes the following source data and figure supplement(s) for figure 1:

**Source data 1.** The labels of 34 ROIs in the ventral cortex.

**Figure supplement 1.** The 34 ROIs of the ventral cortex in the adult and neonatal space.

**Figure supplement 2.** Spatial variation of cortical structure development in the ventral cortex.

**Figure supplement 3.** Example segmentations of subjects scanned at 37–44 weeks.

**Figure supplement 4.** Comparison of the area V1 created by manual and registered methods.

functional and structural development in area V1. Finally, we evaluated the structural and functional differences of visual cortex between the term- and preterm-born babies as the latter group had shorter GA at birth but longer PT compared to the former group with equivalent PMA.

## Results

### General development of cortical thickness and cortical myelination in human infants

To show the general developmental trajectories of CT and CM in the visual cortex during the human neonatal period, we included 407 neonates (187 females, PMA = 41.11 ± 1.73 weeks at scan) from a total dataset of 783 subjects after excluding data that did not pass quality control or did not satisfy inclusion criteria ('Materials and methods'). In total, 355 of them were term-born (PMA = 41.14 ± 1.70, range from 37.43 to 44.71 weeks at scan; *Figure 1a*). The general trend and spatial variation of visual cortex development were described within an anatomical mask of ventral cortex, which was then segmented into 34 regions of interest (ROIs) per hemisphere according to the HCP-MMP atlas (*Glasser et al., 2016*), including the early visual cortex (e.g., V1 and V2), higher-level visual cortex (e.g., VOTC), and anterior part of temporal cortex (*Figure 1—figure supplement 1*, *Figure 1—source data*

*1*). In general, the CT and CM of the ventral cortex significantly increased between 37 and 45 weeks of PMA ($r = 0.40$–$0.49$, $p<10^{-9}$; *Figure 1b–d*). In addition, we found distinct spatial variation along ventral cortex, for example, posterior–anterior and medial–lateral directions (*Figure 1—figure supplement 2a and b*). Generally, both CT and CM showed higher correlation with PMA in the posterior than anterior region ($r = -0.8$ and $-0.83$; $p<0.001$), and higher correlation in the medial than lateral part within the ventral visual cortex ($r = 0.7$ and $0.91$; $p<0.001$; *Figure 1—figure supplement 2c and d*).

## Contribution of postnatal time and gestational age on the development of cortical thickness and cortical myelination

Similar developmental trajectories of CT and CM in the ventral cortex across PMA were observed in the above results. It would be interesting to understand the differential influences of GA (the time from fertilization to birth) and PT (the time from birth to MRI scan) on cortical development because they reflected two very factors (e.g., innate growth versus postnatal experience). The GA of our study population was 39.93 ± 1.26 weeks and the PT was 1.21 ± 1.25 weeks, and the correlation between them was not significant ($r = -0.08$, $p>0.1$; *Figure 2—figure supplement 1*). The results showed that the GA was significantly related to the CM ($r = 0.53$ and $0.51$ for right and left hemisphere, respectively; $p<10^{-9}$) and CT ($r = 0.16$ and $0.14$; $p<0.05$) averaged in the whole ventral mask. In terms of regional correlation, the CM was significantly correlated with GA in all 68 ROIs ($r > 0.28$, false discovery rate [FDR]-corrected $q < 0.05$), while CT showed significant correlation only in 23 ROIs ($r > 0.13$, FDR-corrected $q < 0.05$) (*Figure 2a and b*). The PT was significantly correlated with CT ($r = 0.53$ and $0.51$ for left and right ventral cortex, $p<10^{-9}$) for left and right ventral cortex, with 61 of the 68 ROIs showing significant positive correlation (all $r > 0.11$, FDR-corrected $q < 0.05$). In contrast, the correlation between PT and CM was overall insignificant in the ventral cortex ($r = 0.02$ and $0.01$, $p>0.6$) with only two ROIs in the anterior temporal areas showing significant effects (*Figure 2c and d*). We applied a linear mixed-effect model to test whether the CT (or CM) of the whole ventral cortex was differently influenced by the GA versus PT and found that the GA had a significantly stronger effect on the CM than PT (interaction between GA and PT, $p<0.05$) but no significant difference was found on the development of CT ($p>0.6$).

Furthermore, two subsamples of the dataset were selected to control the effect of one age on the other one. The first subsample included infants who underwent the scans within 3 days after birth (n = 173) to control for PT, and the second subsample included infants whose GA was within a short range from 39 to 40 weeks (n = 100) to control for GA. The patterns in the two subsets were similar to the above results (*Figure 2e–h*). Taken together, those results suggested that the development of CM heavily depends on the endogenous mechanisms and longer prenatal time would be beneficial for the development of myelination. In contrast, the development of CT in neonates was driven by both postnatal and prenatal time.

Next, we looked into the spatial variability within the ventral cortex and focused on two typical visual areas including the V1 and VOTC (15 ROIs) based on HCP-MMP atlas (*Figure 3a*, *Figure 1—figure supplement 1*, *Figure 1—source data 1*). Both regions showed general increase of CT and CM with PMA ($r = 0.23$–$0.61$, $p<10^{-6}$) but were differently influenced by GA and PT. Particularly, the development of CT was significantly modulated by both the PT ($r = 0.26$–$0.52$, $p<10^{-6}$) and GA ($r = 0.18$–$0.29$, $p<10^{-3}$; except for the left VOTC, $r = 0.01$; *Figure 3a–c*) for both regions. But the development of CM was significantly influenced by GA ($r = 0.43$–$0.51$, $p<10^{-6}$) but not PT ($r = 0.02$–$0.09$, $p>0.05$; except for the left V1, $r = 0.12$, $p=0.03$). Moreover, we applied a linear mixed-effect model to test the developmental difference of the cortical structure between GA versus PT. The results showed that the CT in two regions had nonsignificantly different influences between GA and PT ($p>0.3$), but CM showed a significantly different impact from the two factors in V1 ($p<0.01$) and marginal significance in VOTC ($p<0.09$).

## Innate functional organization of ventral cortex in human infants

Beyond structural development of the neonatal cortex, we further asked how the functional connectivity between the different visual subregions changes during early development. To do so, we first estimated the initial functional connectivity pattern in ventral cortex without the influence of postnatal visual experience in a subsample of subjects who underwent the scans within the first day after birth (n = 73). Homotopic correlation was calculated to reflect the functional distinction in neonatal ventral

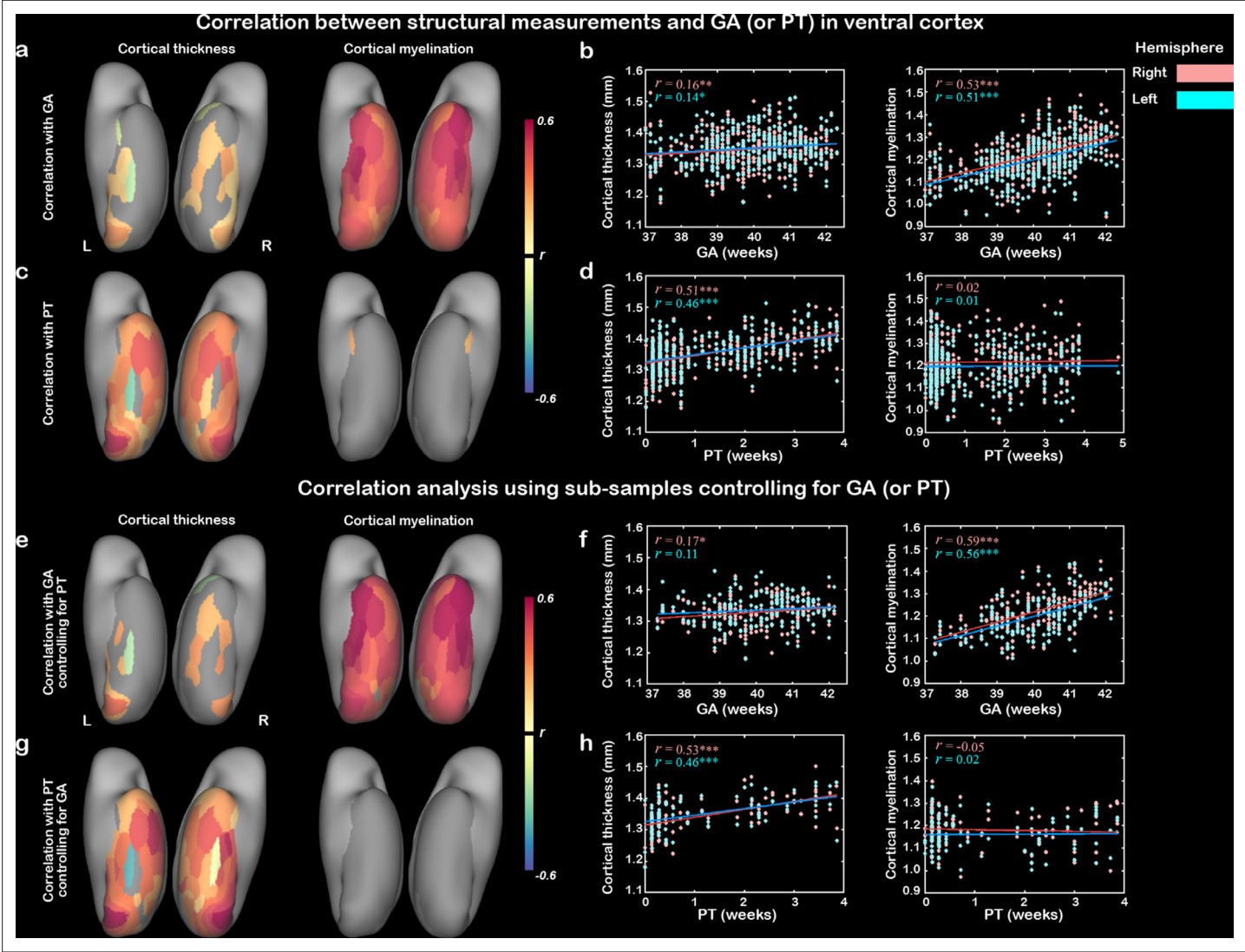

**Figure 2.** Contribution of postnatal time and gestational age on the development of cortical thickness and cortical myelination. The correlation maps between cortical thickness (CT)/cortical myelination (CM) and gestational age (GA) (**a, b**) or postnatal time (PT) (**c, d**) in ventral cortex. (**e–h**) Validation analysis using two subsamples: the first subsample included infants who underwent the scans within 3 days after birth to control for the postnatal time (**e, f**) and the second subsample included infants whose GA ranged from 39 to 40 weeks to control for the prenatal time (**g, h**). Note: correlation coefficients in the nongray areas were significant after false discovery rate (FDR) correlation (FDR $q < 0.05$); * $p<0.05$, ** $p<0.05$, *** $p<0.001$ by Pearson correlation.

The online version of this article includes the following figure supplement(s) for figure 2:

**Figure supplement 1.** Relationship between gestational age (GA) and postnatal time (PT).

**Figure supplement 2.** The correlation coefficient maps between postmenstrual age (PMA) or postnatal time (PT) and the structural measurements (cortical thickness [CT] or cortical myelination [CM]) across the whole brain.

cortex (**Arcaro and Livingstone, 2017**; **Vincent et al., 2007**). The homotopic connections in all ROIs of ventral cortex were significant (mean $r = 0.13–0.43$, $t > 12.87$, $p<10^{-9}$; **Figure 4a and b**) and were significantly higher than adjacent connections ($0.29 \pm 0.12$ vs. $0.19 \pm 0.10$, Wilcoxon signed-rank test on the Fisher-Z-transformed $r$-value: $z = 16.32$, $p<10^{-9}$) and distal connections ($0.04 \pm 0.06$, $z = 16.32$, $p<10^{-9}$; **Figure 4c**), suggesting that the arealization (homotopic connection) of ventral cortex already existed at birth.

In addition, multidimensional scaling (MDS) analysis of the correlations between 34 ROIs within the right ventral cortex revealed the functional relationship among those areas in neonates within 1 day of life (**Figure 4d**). Using community structure analysis on the network matrix (threshold $r =$

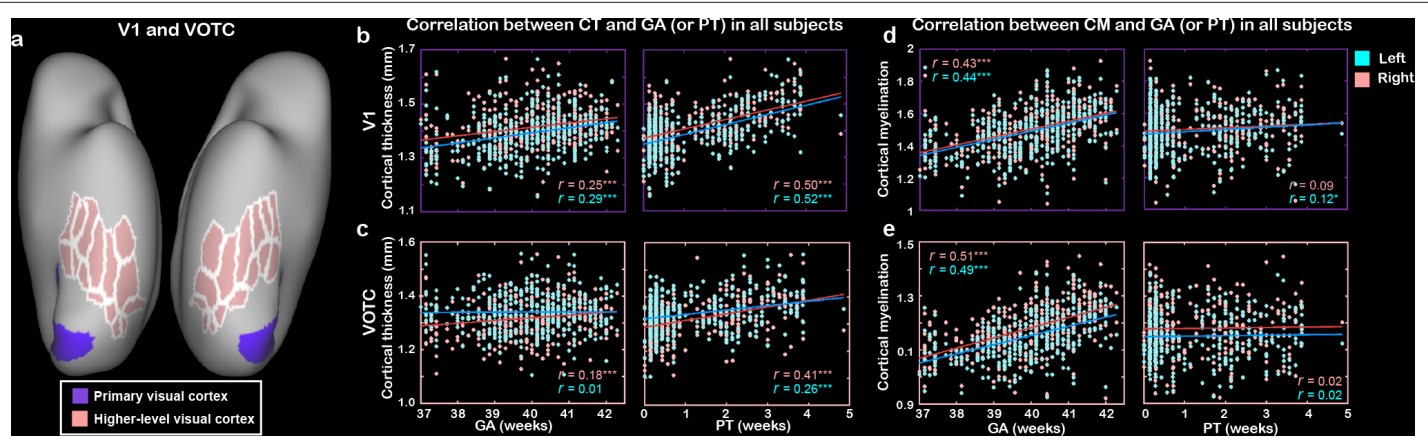

**Figure 3.** The development of cortical thickness (CT) and cortical myelination (CM) in primary visual cortex (V1) and higher-level visual cortex (VOTC) in human newborns. (**a**) Definitions of V1 and VOTC. Correlation between CT (**b, c**) or CM (**d, e**) and gestational age (GA) or postnatal time (PT) in V1 and VOTC. *p<0.05, ***p<0.001.

0.15, p<10$^{-8}$; **Arcaro and Livingstone, 2017**), we further partitioned these areas into three groups, including a lateral cluster, a medial cluster, and an anterior–lateral temporal cluster (**Figure 4d**). Similar results were found with different threshold (e.g., r = 0.10, 0.15, and 0.20) and in the left hemisphere (**Figure 4—figure supplement 1**). Taken together, those results suggested that the proto-organization of ventral visual cortex was formed before acquiring any higher-level visual experience.

## Development of functional connections in the ventral visual cortex

Based on the baseline functional connections in term-born neonates, we were intrigued to know how those functional properties of the ventral visual cortex develop during neonatal stage of human infants? If so, whether PT and GA had different influences on this process? For the arealization in the 34 ROIs, V1 and lateral–anterior regions showed a significant increase with PMA (16/34 regions, r = 0.12–0.27; FDR q < 0.05; **Figure 4e**), suggesting a general development of homotopic connection in the ventral cortex. However, the GA showed significant correlations with the homotopic connections in the anterior temporal cortex (14/34 patches, r = 0.12–0.28; FDR q < 0.05; **Figure 4f**) but PT was significantly related to the connections in V1, fusiform, and anterior temporal areas (6/34 patches, r = 0.14–0.22; FDR q < 0.05; **Figure 4g**). For the entire ventral cortex, we applied MDS and community structure analysis in each PMA week from 38 to 44 weeks across all term-born infants (n = 355) and found similar three-cluster network structures in the infants with different PMA (except for 42 weeks; **Figure 4—figure supplement 2**). Then we used two typical network measurements, including global efficiency and mean cluster coefficient, to quantify the development of the functional network in ventral cortex. Both two measurements showed significant increase with PMA (r = 0.25 and 0.27, p<0.001) and significant correlations with both GA (r = 0.11 and 0.13, p<0.05) and PT (r = 0.22–0.25, p<0.001; **Figure 4—figure supplement 3**).

The correlations between ipsilateral connections and PMA showed both positive and negative results in different regions of the ventral cortex. Particularly, the early visual cortex (e.g., V1 and V2) showed decreasing connection to other visual areas across PMA while some areas in the VOTC (e.g., fusiform and parahippocampus) showed increasing tendency (**Figure 4—figure supplement 4**). Here, we focused on the early connections within the V1 and VOTC (**Figure 5a**) and found the homotopic correlation between contralateral V1's (r = 0.43 ± 0.25) was significantly higher than the correlation between VOTCs (r = 0.38 ± 0.22; t = 4.29, p<10$^{-4}$), which were both higher than the connections between V1-VOTC within the same hemisphere (r = 0.16 ± 0.19; t = 15.74 and 13.49, p<10$^{-9}$; **Figure 5b**) in the term-born infants. Furthermore, the homotopic correlations of both V1 and VOTC showed significant increase with PMA (r = 0.16 and 0.18, p<0.01) but the connection between V1 and VOTC showed significant decrease with PMA (r = –0.23, p<10$^{-4}$). Moreover, the homotopic correlations of V1 and VOTC were significantly modulated by the PT (r = –0.20 and

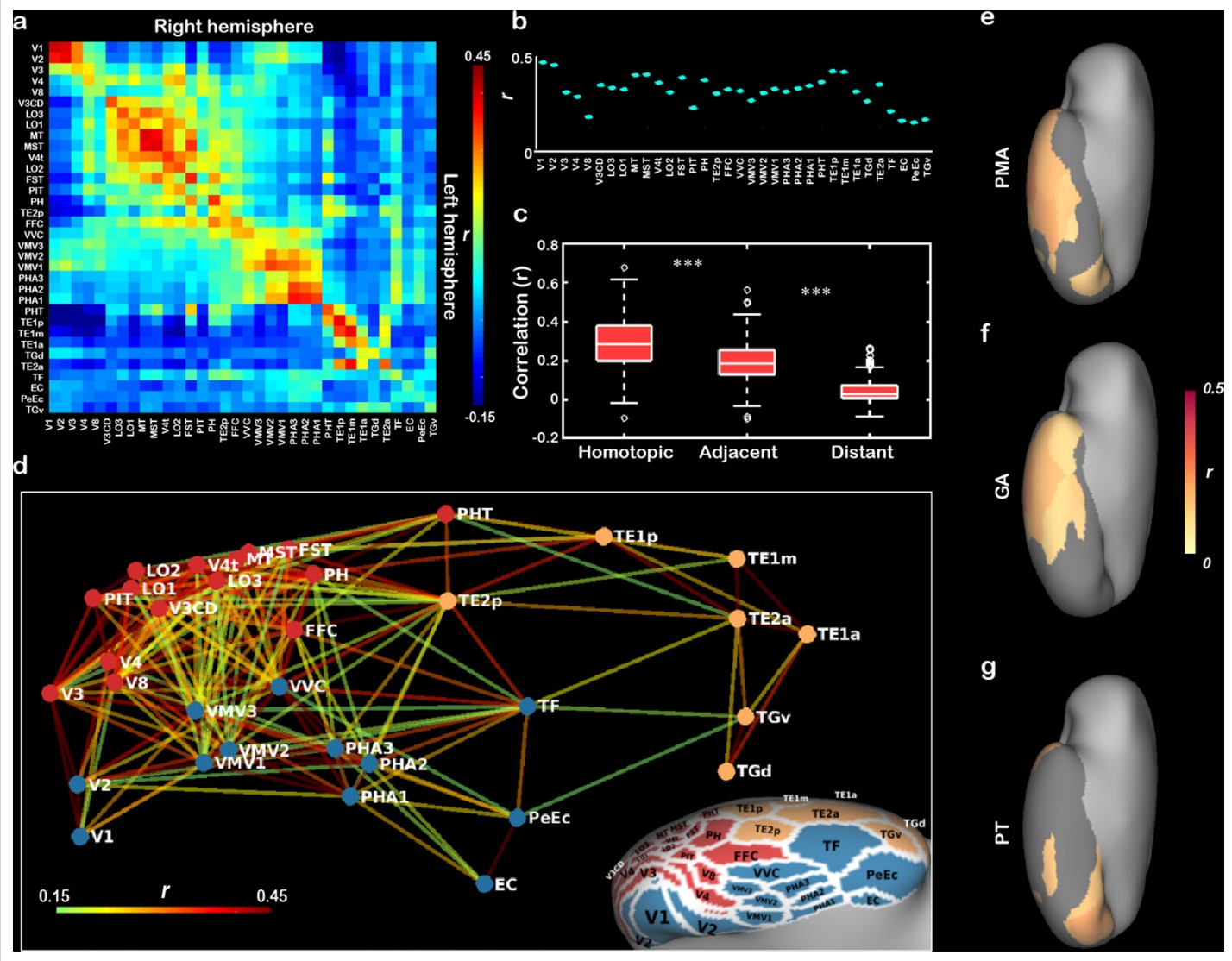

**Figure 4.** The innate functional organization of ventral cortex within 1 day after birth and its development in the first month of age in human newborns. (**a**) The pairwise correlation matrix describes the functional correlations among 34 regions of interest (ROIs) across hemispheres in ventral cortex. (**b**) The scatter graph illustrates the correlation coefficients (*r*) between 34 pairs of bilateral homotopic areas. (**c**) The box graph depicts the comparison between homotopic, adjacent, and distant connections at birth. (**d**) Multidimensional scaling and community groups obtained from the pairwise connections between 34 ipsilateral ROIs in right ventral cortex and the corresponding projection on the cortical surface. The color of the nodes in (**d**) indicates the cluster identity in the community structure analysis, and the color of the lines connecting the nodes indicates the functional correlation (*r* value) between two nodes. The correlation coefficient maps between the homotopic correlation and postmenstrual age (**e**), gestational age, (**f**) or postnatal time (**g**) in 34 ROIs are mapped onto the cortical surface, and the nongray areas indicated the significant correlations after false discovery rate (FDR) correlation (FDR *q* < 0.05). ***p<0.001.

The online version of this article includes the following figure supplement(s) for figure 4:

**Figure supplement 1.** Multidimensional scaling and community groups based on pairwise ipsilateral connections between 34 regions of interest (ROIs) in the left or right ventral cortex.

**Figure supplement 2.** Functional networks of right ventral cortex for each postmenstrual age (PMA) week from 38 to 44 weeks based on multidimensional scaling.

**Figure supplement 3.** The correlation between global efficiency or mean cluster coefficient of the right hemisphere and gestational age (GA) or postnatal time (PT).

**Figure supplement 4.** The correlations between ipsilateral functional connections and postmenstrual age.

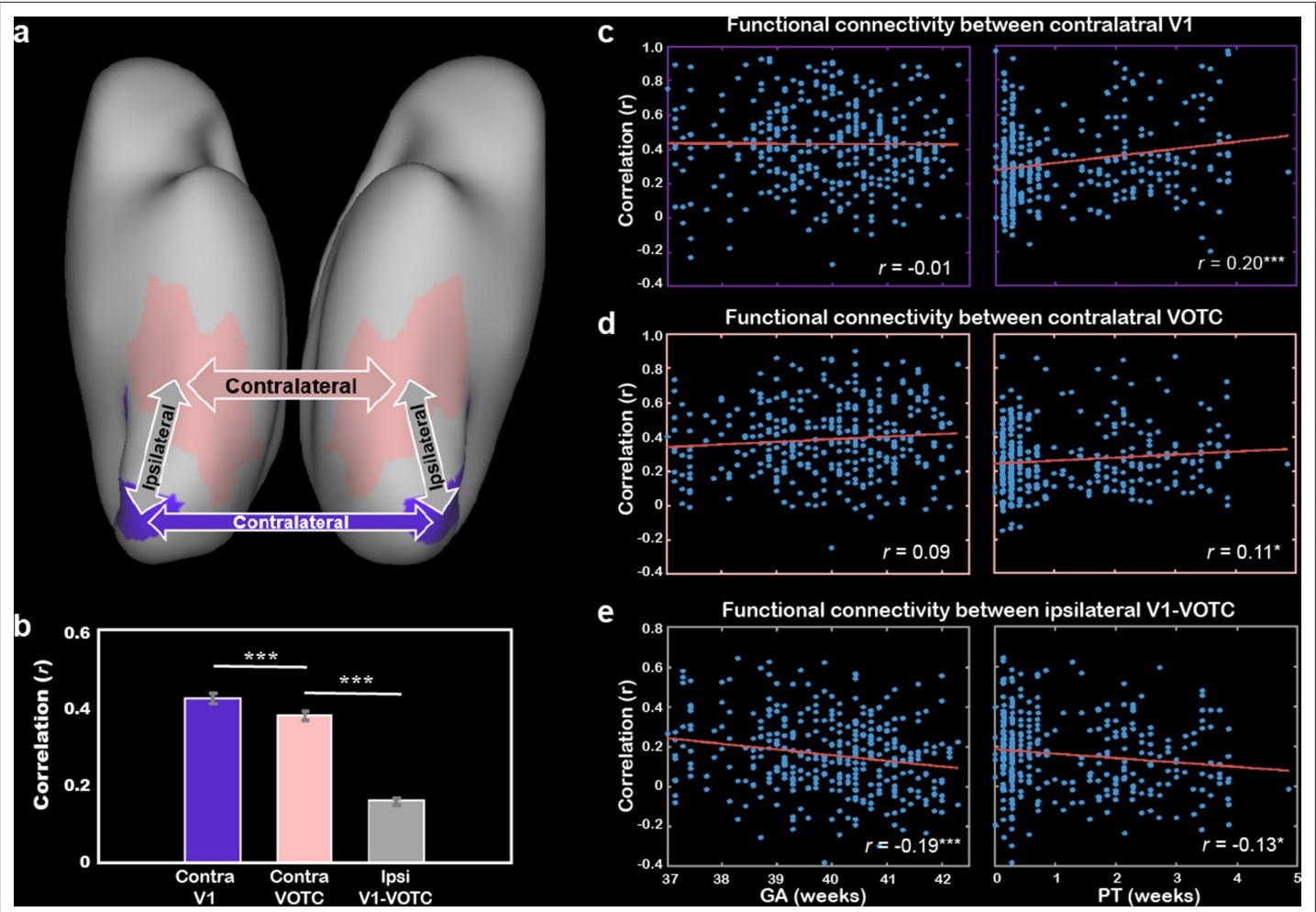

**Figure 5.** The development of the functional connections between bilateral primary visual cortex (V1) and higher-level visual cortex (VOTC) and their cross-correlations. (**a**) The connections of interest include the homotopic connection between bilateral V1 (purple), bilateral VOTC (peach), and the averaged ipsilateral connections between V1 and VOTC in each of the hemispheres (gray). (**b**) Comparison of the three types of connections in the infants within 1 day after birth (no postnatal experience). The correlation between gestational age (GA) or postnatal time (PT) and bilateral V1 connection (**c**), bilateral VOTC connection (**d**), and ipsilateral connection between V1 and VOTC (**e**). The values in the bracket indicate the partial correlation coefficient controlling for postmenstrual age (PMA) of infants. Contra, contralateral; Ipsi, ipsilateral; *p<0.05; **p<0.01, ***p<0.001.

–0.11, p<0.05) but not the GA (r = –0.01 and 0.09, p>0.05; *Figure 5c and d*). The ipsilateral connection between V1 and VOTC was significantly correlated with both PT and GA (r = –0.19 and –0.13, p<0.05; *Figure 5e*).

## Relationship between structural and functional properties in the area V1

The above results revealed that structural and functional properties of the ventral visual cortex both developed with PMA, but were differently influenced by the in utero and external environment (*Table 1*). We further investigated the relationship between structural and functional development in area V1, which showed a strong developmental effect in both structural and functional analyses. Mediation analysis was employed to see whether the development (GA or PT) of the homotopic connection between bilateral V1 was mediated by the structural properties (CT or CM). We found that the PT had a significant direct effect on the homotopic function that was not mediated by CT or CM (*Figure 6a and b*). In contrast, the direct effect of GA on the homotopic connection was not significant but the indirect effect of GA through CM on the connection was significant (*Figure 6c and d*).

**Table 1.** Summary of the effects of gestational age (GA) and postnatal time (PT) on the structural and functional properties of the ventral visual cortex.

The numbers indicate the correlation coefficients of the corresponding measurement and the GA or PT.

| Properties | | GA | PT |
|---|---|---|---|
| CT | Whole ventral cortex | <0.2 | >0.4 |
| | V1 | 0.2–0.4 | >0.4 |
| | VOTC | <0.2 | 0.2–0.4 |
| CM | Whole ventral cortex | >0.4 | Nonsignificant |
| | V1 | >0.4 | <0.2 |
| | VOTC | >0.4 | Nonsignificant |
| FC network properties in the whole ventral cortex | | 0.2–0.4 | <0.2 |
| Homotopic connection | V1 | Nonsignificant | <0.2 |
| | VOTC | Nonsignificant | <0.2 |
| Ipsilateral connection between V1 and VOTC | | <0.2 | <0.2 |

CT, cortical thickness; CM, cortical myelination; VOTC, ventral occipital temporal cortex.

## Comparison of structural and functional features between term- and preterm-born infants

Compared the term-born infants, preterm-born infants have longer PT with relatively shorter GA at birth, and thus the comparison between them may help us to understand the influences of early experience and innate growth on cortical development. Considering the unbalanced sample size between the two groups (n = 355 vs. 52), we selected a subgroup in the term-born infants with equal sample size (n = 52) and similar PMA to the preterm-born neonates (40.92 ± 1.87 vs. 40.90 ± 1.91 weeks; $t = 0.06$, p>0.95) (*Figure 7a*). The term-born infants had significantly higher GA (39.96 ± 1.40 vs. 31.98 ± 3.32 weeks; $t = 14.04$, $p<10^{-9}$) but lower PT (0.96 ± 1.13 vs. 8.91 ± 4.42 weeks; $t = -13.86$, $p<10^{-9}$) compared to preterm-born infants. For the structural measurements, the mean

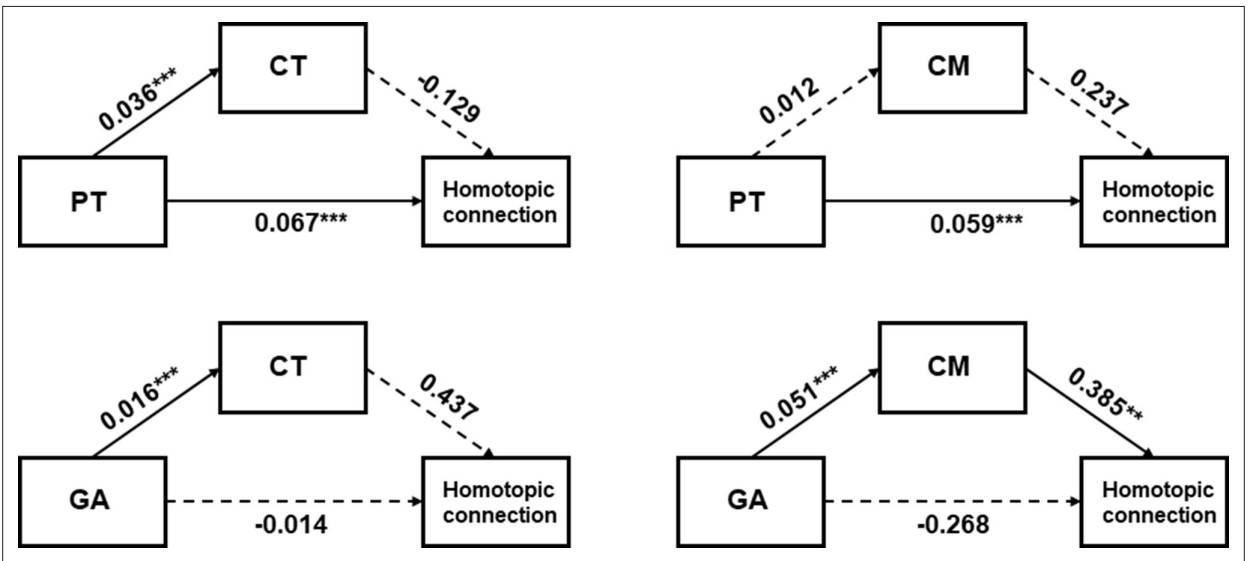

**Figure 6.** Mediation analysis between the developmental factors (gestational age [GA] or postnatal time [PT]), homotopic functional connection between bilateral V1, and structural features (cortical thickness [CT] or cortical myelination [CM]). Homotopic connection between bilateral V1 was set as an independent variate, while the developmental factor was a dependent variate, and the structural features (CT or CM) was the mediated variate. **p<0.01, ***p<0.001.

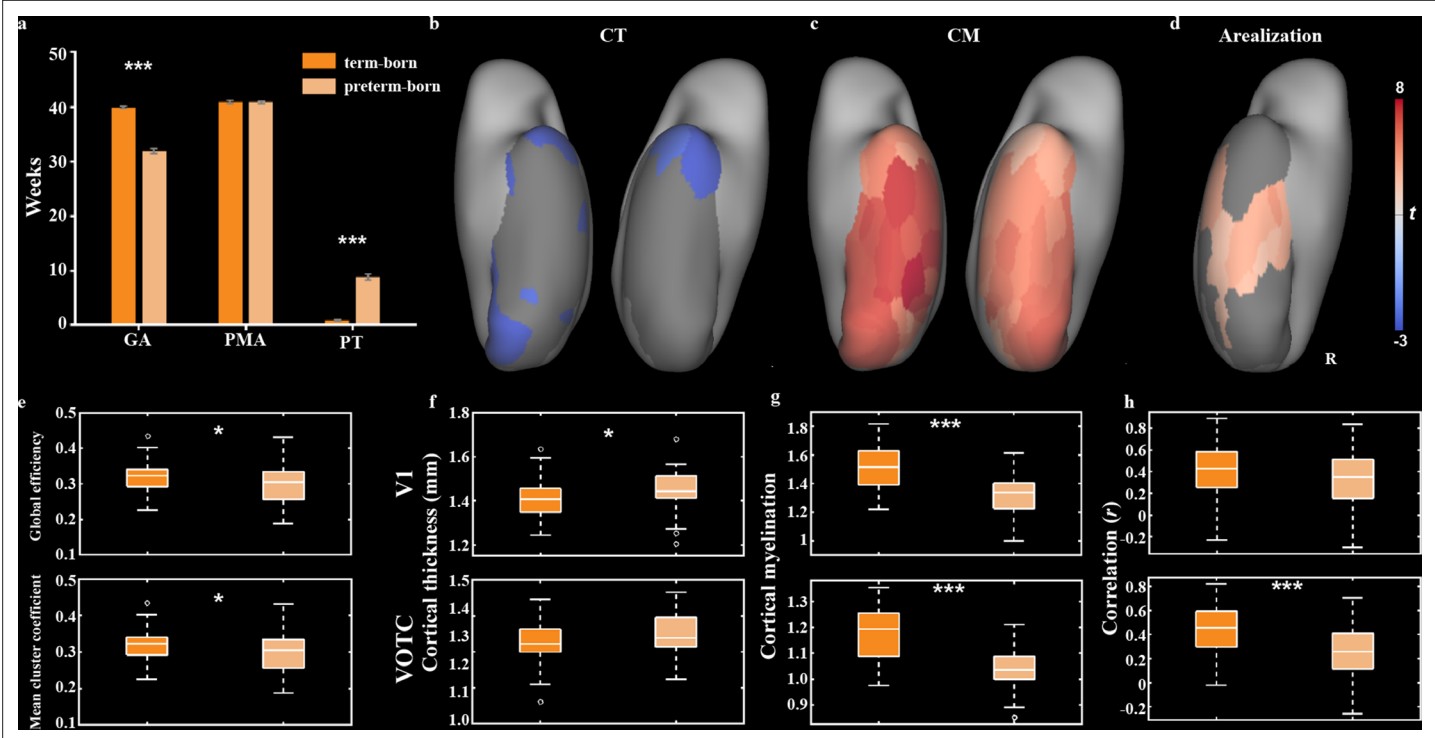

**Figure 7.** Comparison of structural and functional features between term- and preterm-born infants. (**a**) The comparison of gestational age (GA), postmenstrual age (PMA), and postnatal time (PT) between term-born and preterm-born infants. (**b–d**) The *t* maps of cortical thickness (CT), cortical myelination (CM), and arealization between two groups (term – preterm) in ventral visual cortex. Only the significantly different regions (false discovery rate [FDR] *q* < 0.05) are shown. (**e–h**) The box plots of structural and functional measurements between two groups, including the global coefficient and mean cluster coefficient of the ventral network (**e**), CT in V1 and VOTC (**f**), CM in V1 and VOTC (**g**), and functional connectivity between contralateral V1 and VOTC (**h**). *p<0.05, **p<0.01, ***p<0.001 by two-sample *t*-test.

The online version of this article includes the following figure supplement(s) for figure 7:

**Figure supplement 1.** The mean cortical thickness (CT) and mean cortical myelination (CM) in each of the 34 regions of interest (ROIs) in the ventral visual cortex compared between term and preterm-born infants.

**Figure supplement 2.** Functional networks of right ventral cortex in term-born and preterm-born infants, based on multidimensional analysis.

CT across all ROIs was significantly lower in the term-born (1.39 ± 0.05) than preterm-born infants (1.35 ± 0.05; *t* = –3.16, p<0.01) and 10 of 68 ROIs showed significantly lower CT in the term-born than preterm-born infants (*Figure 7—figure supplement 1*). On the contrary, the mean CM across all ROIs was higher in the term-born (1.22 ± 0.10) than preterm-born (1.09 ± 0.08; *t* = 7.11, p<10⁻⁹) infants and all ROIs were significantly higher in the term-born than preterm-born groups. Particularly, the term-born infants showed significantly lower CT in the area V1 (*t* = –2.55, p<0.02) but not VOTC (*t* = –1.24, p>0.2; *Figure 7f*) and higher CM in both area V1 and VOTC (*t* = 6.65 and 7.54, p<10⁻⁸; *Figure 7g*).

For the functional organization of ventral cortex, two groups showed similar three-community structure except that some areas (e.g., V1 and V2; *Figure 7—figure supplement 2*) were included in adjacent community in the preterm-born group. Significantly different network measures were found between two groups, for example, the term-born infants had higher global efficiency and mean cluster coefficient than preterm-born infants (*t* = 2.22 and 2.43, p<0.05; *Figure 6e*), suggesting higher efficiency of information transmission and higher differentiation of cortical function in the ventral cortex of term-born infants. Compared to preterm-born infants, the arealization in the term-born infants was higher in posterior and middle parts of ventral cortex but lower in the anterior temporal areas (*Figure 6d*). Particularly, the preterm-born infants showed lower arealization in the VOTC (*t* = 3.75, p<0.001) but not V1 (*t* = 1.60, p>0.1) than term-born neonates (*Figure 6h*), suggesting that the connections between bilateral visual cortex were also influenced by the GA.

## Discussion

Experience-dependent plasticity is one of the most striking features of human brain and also the foundation of acquired cognitive ability. Utilizing the large datasets of neonatal multimodal MRI images from dHCP, we reported the early structural and functional maps in the ventral visual cortex and their development during the first month of age, focusing on the contributions of prenatal time and PT to the cortical development in this early period. Particularly, we found that PT was significantly related to the cortical thickening and specific functional circuits (e.g., homotopic functional connections between the bilateral visual cortex), while GA was significantly correlated with the CT, CM, and ipsilateral connections in the visual cortex. Such separation reflected the influence of the developmental environment (in utero and outside world) on the cortical development in the infants. Furthermore, we found that the preterm-born infants had higher CT (e.g., V1), lower CM, and lower homotopic connections (e.g., VOTC) compared to term-born infants at equivalent PMA. Taken together, those results suggested that the development of human visual cortex does not strictly follow the two-stage hypothesis (*Barlow, 1975*; *Li et al., 2006*; *White and Fitzpatrick, 2007*) as the cortical properties are modulated by postnatal environment even within the first month of life.

Similar to the previous findings in human infants (*Bozek et al., 2018*; *Fenchel et al., 2020*), both the CT and CM in ventral cortex increased between 37 and 44 weeks of PMA in our study. We went one step further to investigate the different mechanisms underlying these trends and found the development of CT was considerably modulated by both postnatal and prenatal times while the CM was heavily influenced by prenatal duration. CT is related to the synaptogenesis and synaptic pruning processes that are shaped by both innate and postnatal factors. It was reported that human visual cortex thickens to a maximum during the first 2 years and gradually thins thereafter (*Lyall et al., 2015*), mirroring the trajectory of synapses in human striate cortex (*Huttenlocher and de Courten, 1987*). The previous evidence that both congenitally blind and sighted subjects showed uptrend of CT in the early stage, leading to the conclusion that the visual experience might not be involved in this process (*Bourgeois and Rakic, 1996*; *Jiang et al., 2009*). Our results in the human neonates suggested the postnatal environment could accelerate the thickening of neonatal visual cortex, even if it was not necessary.

Is it the visual experience out of various postnatal stimuli that modulated the development of CT in ventral visual cortex? Although we could not provide a direct evidence from this study, there were indications that the CT in this study might reflect an experience-dependent synaptogenesis induced by early visual experience (*Holtmaat and Svoboda, 2009*). On the one hand, we found that across whole brain, PT-dependent increase of CT was most prominent in the V1, primary auditory area, and central sulcus (*Figure 2—figure supplement 2*), which directly receive visual, auditory, sensorimotor stimuli; while the increases of CM were mainly distributed in the frontal areas (*Figure 2—figure supplement 2*). This whole-brain pattern suggested that CT was shaped by postnatal environment universally, and it is reasonable to expect that ventral visual cortex is closely related to visual experience. On the other hand, we found a well-organized spatial pattern of cortical thickening in ventral cortex with the posterior area showing faster development than anterior regions. This might reflect the influence of visual flow along the ventral cortex, for example, the primary visual area in the posterior area processes all kinds of visual information while higher-level visual cortex responds less. Future study could combine the behavioral measurements to test the relation between these two factors or compare the sighted and congenitally blind infants to further illustrate such question.

T1w/T2w-based myelination measurement, which takes advantage of the covariation between cortical myelin content and T1w (positive covariation), as well as T2w (negative covariation) intensities, could be used as a surrogate maker of myelination (*Glasser and Van Essen, 2011*; *Soun et al., 2017*). CM is related to various cognitive performance (*Glasser et al., 2014*), which might reflect the increase of myelinated fibers in the cortex. Previous studies reported the coupling between CM and face processing, suggesting an experience-dependent development of visual cortical microstructure in later life (e.g., 5–12 years old; *Nordt et al., 2021*). However, we found a significant correlation between CM in the ventral cortex and GA but not PT in the neonatal period, suggesting that early myelination was primarily determined by the endogenic growth in the uterine environment. The posterior-to-anterior spatial pattern of CM in the ventral visual cortex agreed with a previous study on infants at around 6 months of age (*Natu et al., 2021*). Our results further suggested that this hierarchical pattern occurs as early as the neonatal period.

Using r-fMRI data, previous studies evaluated the functional organization of the visual system in neonatal macaques (*Arcaro and Livingstone, 2017*) and category-selective network in human infants (6–57 days) (*Kamps et al., 2020*). Both studies revealed the proto-organization prior to the emergence of functional domain such as face and place areas. Herein we validated those findings in a large sample with infants as early as 1 day of postnatal age, fully controlling the influence of postnatal visual experience. Particularly, we found two distinct visual clusters along the medial–lateral axis, which captured the central–peripheral organization of ventral visual system (*Grill-Spector and Weiner, 2014*; *Hasson et al., 2002*; *Levy et al., 2001*; *Wiesel, 1982*). Apart from the border of two medial–lateral clusters, the lateral area becomes more face-selective while the medial area is specialized in processing scene and buildings later in life (*Nordt et al., 2021*), and thereby our observation suggests the innate scaffold is already established at birth for subsequent experience-dependent modification in ventral visual cortex (*Arcaro et al., 2017*; *Arcaro and Livingstone, 2021*; *Arcaro and Livingstone, 2017*). However, the present homotopic connections in the human neonates were lower than those in neonate macaca mulattas (*Arcaro and Livingstone, 2017*). This difference might relate to the higher motion in human infants, less r-fMRI data in this study, coarser parcellation in the visual cortex used in this work, and the developmental difference between primates and humans in the neonatal period. Furthermore, the connections among the ventral visual cortex have developed during this early stage. Specifically, the homotopic connections between bilateral V1 and between bilateral VOTC both increased with PMA, indicating increased degree of functional distinction (*Arcaro and Livingstone, 2017*; *Vincent et al., 2007*). In contrast, the connection between V1 and VOTC decreased between 37 and 44 weeks of PMA, which might also reflect the development of functional differentiation between adjacent regions in the same hemisphere. More importantly, the connection between bilateral V1 and VOTC was significantly modified by PT, supporting that the early postnatal experience not only influences the local structural features of cortical cortex but also the functional circuits. It is worth noting that the increased homotopic connection can be direct or indirect, for example, the effect might be driven by external regions with enhanced connection to both of the areas (e.g., thalamus).

The preterm-born babies have longer PT but shorter GA compared to full-term infants at the same PMA. The findings that CT in the ventral cortex was generally lower in the term-born than preterm-born infants and that CM showed an opposite trend between the two groups supported the above analysis that CT was preferably influenced by PT while CM was largely dependent on GA during the neonatal period. In terms of functional development, we found the homotopic connections were lower in preterm than in term-born infants, in contrast to the above finding that the homotopic connection increased with PT in both V1 and VOTC. These results might be due to the mediation effect of CM on the connection.

In brief, our results suggested that early cortical development is a mixed outcome of endogenous and experience-dependent development in both cortical structure and function, and postnatal experience could selectively modify the endogenous development of visual cortex during early infancy.

## Limitation

One limitation of this study is the comparison between preterm- and term-born infants did not consider the different visual experience in these infants. The preterm-born neonates may experience very different environment than those of the term-born, for example, the preterm environment can be heavily regulated if they were in a NICU, but we did not have detailed information about the postnatal environment to control for it. Meantime, both GA and PT were different between preterm- and term-born neonates. Then any of the differences between the two groups might have come from the combined effects of GA and PT, and unfortunately, we were not able to separate them in this study. Another concern was the partial-volume effect on the cortical measurements. The changing thickness of cortical ribbon during development may change the degree of partial-volume effect, and thus may affect the CM measurement and may contribute to the myelination difference observed between preterm- and term-born groups.

## Materials and methods

### Participants

A total of 887 datasets (783 neonatal subjects) from dHCP (http://www.developingconnectome.org/) were collected. The dHCP study was approved by the UK Health Research Authority (14/LO/1169), and written consent was obtained from the parents or legal guardians of the study subjects. We excluded the subjects who (1) had high radiology score (above two points) reviewed by specialist perinatal neuroradiologist, which indicated possibly clinical or analytical significance (e.g., punctate lesions or other focal white matter or cortical lesions but not considered to be of clinical significance; https://biomedia.github.io/dHCP-release-notes/structure.html) (n = 218, including two datasets without scores); (2) were sedated during scan (n = 5); (3) were scanned early than 37 weeks (n = 91) of PMA; (4) missing myelin maps or r-fMRI data (n = 152); (5) could not pass the quality control of dHCP preprocessing pipelines for the structural (n = 1) or r-fMRI data (n = 9); and (6) failed the cortical registration pipeline (n = 4). Finally, 407 datasets (407 subjects) were selected in this study, in which 355 were term-born (GA: 39.93 ± 1.26 [37–42.29] weeks; PMA at scan: 41.14 ± 1.7 [37.43–44.71] weeks) and 52 were preterm-born (GA: 31.98 ± 3.35 [23.71–36.86] weeks; PMA at scan: 40.9 ± 1.91 [37–44.29] weeks).

### Data acquisition

All neuroimaging data were acquired at the Evelina Newborn Imaging Centre, Evelina London Children's Hospital, using a 3 T Philips scanner with a newly designed neonatal imaging system including a customized 32-channel phased array head coil, an elaborated positioning device, and a custom-made acoustic hood for infant (*Hughes et al., 2017*). Infants were imaged during natural sleep without sedation. Structural (T1 and T2 weighted image), r-fMRI and diffusion MRI (dMRI) were collected within a single-scan session for each neonate over 63 min. T2-weighted (T2w) and inversion recovery T1-weighted (T1w) multi-slice fast spin-echo images were acquired in sagittal and axial stacks with in-plane resolution of 0.8 × 0.8 mm and slice thickness of 1.6 mm (0.8 mm overlap, except for sagittal T1w that used 0.74 mm). Other parameters were as follows: (1) T1w images were acquired with repetition time (TR) = 4795 ms, echo time (TE) = 8.7 ms, inversion time (TI) = 1740 ms, SENSE factor of 2.27 (axial) and 2.66 (sagittal), and filed of view (FOV) = 145 × 122 × 100 mm; and (2) T2w images were acquired with TR = 12,000 ms, TE = 156 ms, SENSE factor of 2.11 (axial) and 2.58 (sagittal), and FOV = 145 × 145 × 108 mm. The axial and sagittal stacks were integrated using a super-resolution method (*Kuklisova-Murgasova et al., 2012*). High temporal resolution r-fMRI optimized for neonates were collected using echo-planar imaging with a multiband factor of 9 in 15.05 min: TR = 392 ms, TE = 38 ms, 2300 volumes, flip angle = 34°, and spatial resolution = 2.15 mm isotropic (https://biomedia.github.io/dHCP-release-notes/acquire.html; *Bozek et al., 2018*; *Fitzgibbon et al., 2020*).

### Data preprocessing

We collected the preprocessed anatomical and r-fMRI data from dHCP database (*Makropoulos et al., 2018*). For the anatomical data, briefly, preprocessing of dHCP pipeline included super-resolution reconstruction to obtain the 3D T1w/T2w volumes (*Kuklisova-Murgasova et al., 2012*), registration (from T1w to T2w), bias correction, brain extraction, segmentation (on T2w volume using DRAW-EM method, *Makropoulos et al., 2014*; see *Figure 1—figure supplement 3*), surface extraction (*Schuh et al., 2017*), and surface registration (*Robinson et al., 2018*). The initial structural data from dHCP in this study included the individual brain surfaces, CT (corrected version for uneven vertex sampling of gyri relative to sulci), and T1w-/T2w-based myelination maps. Surfaces and cortical metrics of each neonate were nonlinearly aligned to the dHCP symmetric template (40 weeks of PMA; *Bozek et al., 2018*) using multimodal surface matching (MSM, *Robinson et al., 2018*).

The collected functional data from dHCP included the r-fMRI data in individual space and the motion parameters (*Fitzgibbon et al., 2020*). These images were further preprocessed with the following steps using custom codes and DEPABI toolbox (*Yan et al., 2016*) in MATLAB (v2018a). (1) A conservative approach was adopted to address the severe head motion in infants. Similar to the previous study (*Eyre et al., 2021*), we selected a continuous subset (1600 volumes, around 70%) of the data with lowest head motion for each of the infants. Specifically, we first estimated the variation of head motion for each time point within a 50 volume window by calculating the standard deviation of the head motion within the time window, then we calculated the sum of the variations in all possible

1600 continuous volumes (e.g., 1–1600, 2–1601, …, 701–2300), and finally chose the subset with lowest sum of the head motion variation. (2) Registration of the selected r-fMRI subset to individual T2w space with FLIRT (*Jenkinson et al., 2002*). (3) Linear detrending. (4) Regression of nuisance covariates, including 24 head motion parameters and the signals of white matter, cerebrospinal fluid, and global brain. These tissue masks were extracted from the T2w-based segmentation of the corresponding subject. (5) Temporal bandpass filtering with a pass band of 0.01–0.08 Hz. (6) The filtered r-fMRI volume was further projected into the individual cortical surface and then registered to the dHCP symmetric template using MSM method. The resulting surface data was slightly smoothed with 2 mm full-width-half-maximum (FWHM) using workbench (*Glasser et al., 2013*).

## Anatomical ROIs

All ROIs were defined in adult space based on the HCP-MMP atlas (*Glasser et al., 2016*) and registered into the dHCP neonatal template (40 weeks) using MSM method with cortical sulcus as the features to drive the alignment (*Robinson et al., 2018*). We then registered all neonatal surfaces onto the dHCP template using the transformation matrices provided by the dHCP pipeline. We visually checked the registration results for some of the typical sulci and gyri in *Figure 1—figure supplement 1*. Although the best way is to use a neonate-specific parcellation for the present analysis, there is no such fine cortical parcellation of human neonates yet. In addition, we used a definable area – the V1 area – as an example to quantitatively evaluate the registration quality. Specifically, we manually delineated the area V1 on the gradient map (*Figure 1—figure supplement 4a*) of the averaged CM map and calculated the dice coefficient between the transformed V1 from Glasser atlas and the manually delineated V1. The result showed a dice score of 0.83 and 0.80 for right and left hemispheres, respectively. As shown in *Figure 1—figure supplement 4b*, the location of the registered V1 and manually delineated V1 area was consistent, although the latter was slightly bigger. We compared the parcellations on the neonatal atlas to those on an adult the S1200 group-average data collected from the human connectome project (HCP; https://www.humanconnectome.org/) and found the parcellations were visually acceptable at large scale (*Figure 1—figure supplement 1*).

The ventral cortex was parcellated into 34 ROIs per hemisphere in the HCP-MMP atlas, which contain basic visual cortex (e.g., V1 and V2), higher-level visual cortex (e.g., VOTC), and anterior part of ventral temporal cortex (*Figure 1—figure supplement 1*, *Figure 1—source data 1*). The VOTC includes 15 ROIs per hemisphere, including common category-selective regions such as parahippocampus, middle fusiform, and lateral occipital cortex (*Bi et al., 2016*; *Grill-Spector and Weiner, 2014*). The primary visual area was defined as the V1 ROI in the HCP-MMP atlas (*Figure 1—figure supplement 1*).

## Data analysis

### Experimental design

The purpose of this study is to describe the structural and functional development of ventral visual cortex in human newborns and evaluate the influence of postnatal experience in this process. To depict a comprehensive picture of the spatiotemporal development, for each structural or functional measurement, we first presented the general developmental trend of whole ventral cortex and also the spatial variations. Then we focused on the V1 and VOTC and characterized the detailed developmental patterns of these two areas.

To evaluate the contributions of GA and PT on the development, we first used Pearson correlation between the developmental factors and specific structural properties. In addition, we designed two subsamples to validate the independent influence of GA (or PT). The first subsample included infants who underwent the scans within 3 days after birth and thus had limited postnatal experience, and the second subsample included infants within a limited GA range from 39 to 40 weeks to estimate the effect of prenatal time.

Lastly, we compared the term-born infants with preterm-born peers who had longer postnatal experience at equivalent PMA, which may help to explain the influence of early experience versus endogenous coding on cortical development. Therefore, for the structural and functional measurements that were significantly influenced by PT, we will further compare them between term and preterm-born infants.

## Statistical analysis

For all of the correlation analysis, we removed the data beyond 3 SDs from the mean value across all infants. The correlation coefficients were Fisher-Z-transformed before any statistical test, and the plots (e.g., box plots and scatter plots) used the original *r* value to give an intuitive picture. Extraction and calculation of cortical measurement (e.g., CT, CM, and r-fMRI) were performed using gifti toolbox (https://www.nitrc.org/projects/gifti/) in MATLAB (v2018a) and commands in workbench (v1.5.0, https://github.com/Washington-University/workbench; *Harwell, 2022*). Statistical analysis, including Pearson correlation, *t*-test, and linear mixed-effect model, was performed in MATLAB. Mediation analysis was conducted using the PROCESS toolbox (v4.1) in SPSS (v21). FDR correction was used for the multiple-comparison correction if not specifically mentioned. p<0.05 or corrected *q* < 0.05 were considered significant for all tests.

## Structural data

### Cortical thickness and myelination maps of ventral cortex

To generate the average cortical measurements (CT and CM) at the group level, we first registered the metrics of every subject onto a common 40-week template surface using the registration files provided by dHCP pipeline, and then averaged them in a vertex-wise manner across subjects within a specific week (e.g., 38-week map included neonates whose PMA ranged from 37.5 to 38.5 weeks). Because there were only two infants in the 37-week PMA group (37–37.5 weeks) and one infant in the 45-week PMA group (44.5–45.5), we did not present the averaged maps of those weeks.

### Spatial differences between developmental trajectories

To quantify the spatial difference in the developmental change of these cortical measurements, we extracted the default coordinate system of the ventral cortex in the atlas, in which the y-axis was along the anterior–posterior direction and the x-axis was along the medial–lateral direction (*Figure 1—figure supplement 2b*). For each axis, we divided the ventral cortex into 30 segments with equal length along the axis. We then averaged the correlation coefficients across all vertexes within a segment to obtain the spatial pattern of cortical development along the two axes (*Figure 1—figure supplement 2b–d*).

### Comparison of the developmental pattern of cortical structural properties with respect to GA versus PT

We applied a linear mixed-effect model to test whether structural measurements (CT or CM) were differently influenced by GA versus PT, in which CT or CM was independent variate, GA, PT, and their interaction were fixed effect, and the intercept was the random effect.

## R-fMRI data

### Temporal correlation analysis

For each subject, Pearson correlations were carried out on the ROI-averaged time series within and across the left and right ventral cortex. The resulting connections were divided into three groups, namely, the homotopic connection (the connection between two paired areas in two hemispheres. for example, right and left V1), adjacent connection (e.g., right V1 and left V2 since V1 and V2 are adjacent), and distant connections (two areas that were not the paired or adjacent). Independent-samples *t*-test was used to test whether the homotopic correlation was significantly greater than zero across subjects. To compare the correlation among the three types of connections, we applied a nonparameter statistical analysis (Wilcoxon signed-rank) across subjects.

### Multidimensional scaling

Similar to the method used in the previous studies (*Arcaro and Livingstone, 2017*), for each hemisphere, we obtained the ipsilateral correlation matrix between the 34 patches in ventral cortex. Then the correlation matrix was transformed into a distance matrix using 1 – *r* for each entry of the matrix. Nonclassical MDS was carried out on the distance matrix with Kruskal's normalized criterion in MATLAB. The 34 ROIs were plotted in a two-dimensional coordinate according to the first two principal values in the MDS analysis.

## Clustering analysis

For the correlation matrix between the 34 ROIs within ipsilateral ventral cortex, a threshold of $r = 0.15$ was used to remove weak and negative connections. Regional clusters in the ventral cortical network were obtained using a spectral algorithm (*Newman, 2006*) in the Brain Connectivity Toolbox (BCT; *Rubinov and Sporns, 2010*). Different threshold (e.g., 0.1 and 0.2) was also used to validate the results (*Figure 4—figure supplement 1*).

## Developmental changes of ventral network

Two typical network measurements, including global efficiency and mean cluster coefficient (calculated in BCT), were used to quantify the integration and segregation of the ventral network. Pearson correlation between two measurements and PMA (or PT) was used to describe the development of the ventral network in human newborns.

## Developmental change of the connections in V1 and VOTC

Four ROIs, including V1 and VOTC in two hemispheres, were included in this analysis. Pearson correlation was performed between the averaged time series in each four ROIs, resulting in two contralateral connections between V1 and VOTC, and two ipsilateral connections between V1 and VOTC in each hemisphere. We then calculated the Pearson correlation coefficient between these connections (Fisher-Z-transformed) and GA (or PT) to measure the developmental changes.

## Comparison between term-born and preterm infants

Due to the unbalanced sample size between preterm-born (n = 52) and term-born infants (n = 355), we selected a subgroup in the term-born infants with equal sample size and similar PMA (40.92 ± 1.87 weeks) to the preterm-born neonates (40.90 ± 1.91 weeks; $t = 0.06$, $p>0.95$). Specifically, for each infant in the preterm-born group, we chose a matched term-born subject with the closest PMA. Structural and functional measurements were compared between two groups using two-sample $t$-test.

## Data and code availability

The data used in this work are available online on the dHCP page (http://www.developingconnectome.org/data-release/third-data-release/).

Main MATLAB codes used for data preprocessing and analysis are available at https://github.com/MingyangLeee/Neonatal-Visual-Cortex. (*Li, 2022*; copy archived at swh:1:rev:d9e50dc27b6ae577e644b6da4abc2ec6d68b7900).

## Acknowledgements

This work was supported by the Ministry of Science and Technology of the People's Republic of China (2018YFE0114600, 2021ZD0200202), National Natural Science Foundation of China (81971606, 82122032), and Science and Technology Department of Zhejiang Province (202006140, 2022C03057). Data were provided by the developing Human Connectome Project, KCL-Imperial-Oxford Consortium funded by the European Research Council under the European Union Seventh Framework Programme (FP/2007–2013)/ERC Grant Agreement no. 319456. We are grateful to the families who generously supported this trial.

## Additional information

### Funding

| Funder | Grant reference number | Author |
| --- | --- | --- |
| The Ministry of Science and Technology of the People's Republic of China | 2018YFE0114600 | Dan Wu |
| National Natural Science Foundation of China | 81971606 | Dan Wu |

| Funder | Grant reference number | Author |
|---|---|---|
| Science and Technology Department of Zhejiang Province | 202006140 | Dan Wu |
| The Ministry of Science and Technology of the People's Republic of China | 2021ZD0200202 | Dan Wu |
| National Natural Science Foundation of China | 82122032 | Dan Wu |
| Science and Technology Department of Zhejiang Province | 2022C03057 | Dan Wu |

The funders had no role in study design, data collection and interpretation, or the decision to submit the work for publication.

## Author contributions

Mingyang Li, Conceptualization, Data curation, Software, Formal analysis, Validation, Investigation, Visualization, Methodology, Writing - original draft, Writing - review and editing; Tingting Liu, Data curation, Software, Validation, Investigation, Methodology; Xinyi Xu, Software, Validation, Visualization, Methodology; Qingqing Wen, Visualization, Methodology; Zhiyong Zhao, Software, Methodology; Xixi Dang, Conceptualization, Investigation; Yi Zhang, Resources, Data curation; Dan Wu, Conceptualization, Resources, Data curation, Supervision, Funding acquisition, Investigation, Visualization, Methodology, Project administration, Writing - review and editing

## Author ORCIDs

Mingyang Li ⬤ http://orcid.org/0000-0002-1639-9891
Dan Wu ⬤ http://orcid.org/0000-0002-9303-5821

## Ethics

The dHCP study was approved by the UK Health Research Authority (14/LO/1169) and written consent was obtained from the parents or legal guardians of the study subjects.

## Decision letter and Author response

Decision letter https://doi.org/10.7554/eLife.78733.sa1
Author response https://doi.org/10.7554/eLife.78733.sa2

# Additional files

## Supplementary files

• MDAR checklist

## Data availability

The data used in this work are available online on the dHCP page (http://www.developingconnectome.org/data-release/third-data-release/).

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

# Appendix 1

**Appendix 1—key resources table**

| Reagent type (species) or resource | Designation | Source or reference | Identifiers | Additional information |
|---|---|---|---|---|
| Software, algorithm | MATLAB | MathWorks | RRID:SCR_001622 | |
| Software, algorithm | SPM | FIL | RRID:SCR_007037 | |
| Software, algorithm | DPABI | doi:10.1007/s12021-016-9299-4 | RRID:SCR_010501 | |
| Software, algorithm | FSL | FMRIB | RRID:SCR_002823 | |
| Software, algorithm | MSM | doi:10.1016/j.neuroimage.2017.10.037 | | |
| Software, algorithm | BCT | doi:10.1016/j.neuroimage.2009.10.003 | RRID:SCR_004841 | |
| Software, algorithm | Connectome Workbench | Connectome Workbench | RRID:SCR_008750 | |

