## [Editor Report]

We believe that this study will make significant contributions to developmental neuroscience and vision science as it is a novel attempt to study the processes that might be innate or genetically wired and those that emerge due to worldly experiences within the sensory systems. The authors suggest that early postnatal experience and time spent inside the womb differentially shape the structural and functional development of the visual cortex. The use of large neonatal dataset from the developmental Human Connectome Project is impressive and strengthens the claims made in the paper.

---

## [Decision Letter]

**Decision letter after peer review:**

Thank you for submitting your article "Development of visual cortex in human neonates are selectively modified by postnatal experience" for consideration by *eLife*. Your article has been reviewed by 3 peer reviewers, one of whom is a member of our Board of Reviewing Editors, and the evaluation has been overseen by Joshua Gold as the Senior Editor. The following individual involved in the review of your submission has agreed to reveal their identity: Cameron Ellis (Reviewer #2).

Essential revisions:

Overall, the assessment of the 3 reviewers is highly consistent. They believe that this study will make significant contributions to developmental neuroscience and be of broad interest to the developmental neuroscience community. That being said, the reviewers brought up a number of questions and concerns that they would like for you to address before the manuscript can be further evaluated. The reviewers all provided detailed recommendations that you should consider for your revision (refer to "Recommendations for the authors" for individual reviewer clarifications), but here are the main points that we invite you to address in the revision. Naturally, at this point, there is no guarantee that the revision will be accepted for publication. This depends on an evaluation of your responses by the reviewers, as well as any new issues that arise during the process.

Major Concerns:

1. One important concern brought up by all three reviewers is the confusion with the terminologies GA, PMA, and post-natal time (PT), for preemies/full-term infants. Please define and simplify terms as suggested by the reviewers to ameliorate the readability and justify why the authors chose to use each of the terms at different times in the analyses. Overall, the reviewers suggested using GA and PT and replacing the analyses with GA to simplify the findings. Please see individual reviewers' comments to simplify the language and avoid confusion across the manuscript.

2. To make the results and the flow of the paper more cohesive, we recommend the authors frame and justify each analysis with a clear question that the analysis will answer or the hypothesis that the authors posit as it is often difficult to understand how an analysis being described would contribute to the overall claims of the paper.

3. Relatedly, the reviewers also brought up important points in several places whereby the claims made in the paper are not supported by the analyses conducted under this claim (e.g.., in cortical myelination and overall functional connectivity analyses in Figure 2,4, and 5 as well as the final results). The statistical tests and analysis conducted need to support the conclusions and claims. We recommend that the authors rephrase or tone down the conclusions made in this analysis or reassess their analyses to support the claims. A similar concern is raised by reviewers 1 and 3 in the final section "Comparison between structural and functional properties…", as the reader is left expecting a formal comparison between structural and functional development. Please consider an analysis as suggested by reviewer 3 for e.g., a regression with functional connectivity and structural metrics, to ask for example if a region's homotopic connectivity is correlated with CT or CM and which aspect of time (gestation or postnatal) is most important.

4. Given the authors used an adult atlas to draw regions on the infants' brains, a sanity check of how well the Glasser atlas regions align on a neonatal brain at e.g., 37 weeks and compare that to 42 weeks old and adult brain and running some statistics on how these cortical-based alignments from the adult brain to the neonatal brain align, for a few sample subjects is recommended.

5. A similar sanity check is also essential to check the white-gray matter boundary segmentation in some sample infant data as the authors use cortical thickness as one of their measures and CT is measured as the distance between the pial surface and gray-white matter boundary and the cortical ribbon of infants is thin at birth, there may be a possibility that partial-volume effects could be more prevalent in less-developed infants and impact myelin metrics.

6. The reviewers also asked if the authors used Fisher Z to transform their correlations? This is important to clarify as a significant portion of the results in the paper are correlations and it is unclear from the methods whether the authors are transforming their correlation values to make that use appropriate.

7. Since one of the big conclusions of this paper is that not all structural and functional aspects are affected by gestation or postnatal time the same way, having a quick summary of those findings would be helpful. The reviewers suggest a summary table showing the morphological and myelin measures against gestational time and prenatal time, whether the gestational or prenatal time was a significant modulator of that measure.

8. Reviewer 2 brings up an important point about the homotopy analysis that needs clarification and perhaps rethinking of the statistical tests (e.g., comparing 34 homotopic pairs with the hundreds of non-homotopic pairs) that must be addressed in the revision. Please also consider the recommended alternative test by reviewer 2 that would compensate for some of the constraints to measure the homotopic correlation for a region.

*Reviewer #1 (Recommendations for the authors):*

While I find this work theoretically well-motivated and the use of the large dHCP dataset very exciting, there are some major concerns that must be addressed before this study is ready for publication:

1. There was a lot of confusion for me about the terminology between GA, PMA, and post-natal time, for preemies/full-term infants. To simplify the matter, I would first define these ages for the reader and also choose to use a single term, preferably GA as postmenstrual age would be GA plus the chronological age. This would make it easy to follow the flow of the analysis, especially in the section: "Contribution of postnatal time on the development of cortical thickness and cortical myelination". I also feel the same about the analysis in figure 3, it would make it clearer to use GA and postnatal time, as PMA already included both and it's difficult to see the relationship of the GA on CT and CM separately.

2. Relatedly, I really like Supplementary Figure 4c and 4f, and they may be your point on the postnatal age and GA influence on CT and CM really clear and they can be moved as one of the main figures in the paper.

3. It would be good to sanity check how the adult Glasser maps align on a neonatal brain at e.g., 37 weeks and compare that to 42 weeks old and adult and run some statistics on how these cortical-based alignments from the adult brain to the neonatal brain align, for a few sample subjects. Another sanity check that is necessary especially with the cortical thickness measure is to check the segmentations in the data. It would be useful to provide a montage of a few sample subjects showing the segmentation of the gray-white boundary as the contrast properties in infants is hard to decipher.

4. In the first analysis are there any hemispheric differences in the relationship between CT and CM and age?

5. Can the authors describe how the average cortical thickness and myelination maps in Figures1 b, c are generated? Are the individual baby brains aligned to a common surface? Or is it an average of each of the 34 rois, averaged across the babies and projected on a sample surface?

6. I really liked the functional connectivity analysis, however, I am unclear what the non-homotopic means in this case. Can the authors also explicate what negative connectivity would mean and if it is significant what it would imply? It is also unclear how a correlation of r=.09 in Figure 4b would be significant and passes a multiple comparison threshold. Please clarify.

7. In "Comparison between structural and functional properties in ventral visual cortex of term- and preterm-born infants" analysis: The juxtaposition of these two statements is very confusing to me: "For the structural measurements, 56 of 68 ROIs showed significant lower CT in the term-born than preterm-born infants (ts = -4.26 to -2.71; FDR q < 0.05; Figure 6b). Particularly, the term born infants showed significantly higher CT in the area V1 (t = -2.55, p < 0.02) but not VOTC (t = -1.24, p > 0.2; Figure 6f)." I would rather see a mean CT (mean/std) across the rois in numbers and how it varies between preterm and term-born infants.

8. Relatedly in figure 6b-c, it is difficult to know the significant ts. Please change the non-significant ts to a different color or grays.

9. The analysis titled "Comparison between structural and functional properties in ventral visual cortex of term- and preterm-born infants" is a bit confusing to me. I am not clear on how we learn anything about the structural-functional relationship. It would be more convincing to see some comparative analysis between the structural and functional measures and how this varies by age, and in pre and full-term infants. The current analysis is not doing justice to the title of this analysis.

10. Your results of more myelination in the V1 than higher-order areas are consistent with the hierarchical findings of a prior paper on infants in their first six months of life (see Natu, et al., 2021, Nature Communications Biology). What is interesting to see is that neural scaffolding starts to develop early on during prenatal life. However, it is also interesting that you found that the cortical myelination in the ventral cortex was not directly influenced by postnatal time and I think these points can be discussed in the discussion.

11. Finally, I am not entirely clear if the term postnatal "experience" used throughout the manuscript, is appropriate in this case, perhaps the authors can tone it down the postnatal time, as we don't have any behavioral measure of visual experience per se, like contrast properties, acuity, etc.

*Reviewer #2 (Recommendations for the authors):*

Most of my recommendations are integrated into the public comments. That said, I wish to emphasize how important I think this question is and the impressive amount of work the authors have already done.

A general style comment is that I had trouble following the thread in the results. I think that each analysis should be framed more clearly in terms of the question it is answering. It was often difficult to understand how an analysis being described would contribute to the overall claims of the paper and what alternative hypotheses were possible for the analysis. This led to what felt like a list of results, rather than a coherent narrative.

Recommended analyses:

Regarding point 3: An alternative to this analysis I think the authors ought to consider is to control PT (i.e., find preterm and term groups that have an equivalent postnatal age) and see how variability in GA contributes to the metrics they observe. This reverses the logic of the analysis the authors did, but does so in a way I think is tightly controlled.

Regarding point 9: an alternative test that compensates for some of these constraints is to get the homotopic correlation for a region and then subtract it from the average of the non-homotopic correlations for that region (could even just use the distal pairs for this). This difference can then be compared across individuals for each region.

Regarding point 10: the authors should consider using the Wang et al., (2015) atlas, since that more carefully divides visual cortex.

*Reviewer #3 (Recommendations for the authors):*

– The language at times gets confusing referring to things such as PMA, GA, PA, etc. So I would simplify and clarify the language and say "gestational time" or "gestational age" when referring to how long a neonate spent gestating and "postnatal time" when referring to how much experience it has had after birth. Gestational time and postnatal time are not that long character-wise and will go a long way in improving the readability of the manuscript. I found myself having to go back and look up acronym meanings several times.

– In figure 2, the outline is hard to see in some panels. For example, in Figure 2a, it at first looks like only that central fusiform ROI is the only ROI with a significant value, but I didn't realize until later that it's actually all of the other ROIs that seem to be outlined and the fusiform ROI is not significant. I would say that if an ROI is not significant, I would just black it out, that way if an ROI is colored it is because that value is significant. This will be more straightforward than the current outlines.

– Supp Figure 4 seems important why isn't it a main figure or part of main figure 3? 3 could be condensed or pruned to make room for it. Also on the topic of supplementary figure 4, from what ROI was this taken? All of the visual HCP-MMP ROIs (like V1 + high-level regions) or was it one particular region? If incorporated into Figure 3, showing the data for cortical myelination like parts D and E would be great.

– Figure 4d, would it be worthwhile to make this figure comparing infants with longer PMA times since it was shown that functional connectivity network properties change with development? It might be nice to see how and if the network in Figure 4D changes (do some regions get closer together, etc).

– Typo in figure titles in Figure 5.

– Page 15 Line 3 states that term-born infants have a lower cortical thickness, but then on line 5 it says that term-born infants have a higher cortical thickness in V1. I think this is a typo, Figure 6F would suggest term-born have thinner V1.

– On the topic of Figure 6, it is being shown that the cortex is thinner in V1 in term-born infants who have comparatively less postnatal time. It is also shown that these term-born infants have more myelin. Typically, myelin increases with development, and the cortical thinning that occurs later in childhood is thought to be a result of increasing myelination at the gray-white matter border (see Natu et al. 2019 PNAS). Is it possible some of this myelin difference could result from the fact that in term-born infants the cortex is thinner (1.4mm in V1) and thus some voxels could partial-volume white matter voxels more easily? I know that white matter in neonates appears darker than the cortex in a T1-weighted image, but just want to bring up this potential point that a thinner cortex, given constant-sized voxels, is more likely to be biased by white matter signal from voxels sampling deep cortical layers.

– It might be nice to have a summary table showing, for each measure you compared against gestational time and prenatal time, whether the gestational or prenatal time was a significant modulator of that measure. Since one of the big conclusions of this paper is that not all structural and functional aspects are affected by gestation or postnatal time the same way, having a quick summary of those findings would be helpful.

– In the last Results section "Comparison between structural and functional properties…", I was expecting the authors to more formally compare structural and functional development. I was expecting them to do a regression with functional connectivity and structural metrics, to ask for example if a region's homotopic connectivity is correlated with CT or CM and which aspect of time (gestation or postnatal) is most important. You could even model gestational time and postnatal time separately. I say this because the data currently in Figure 6 while certainly useful, are summaries of data already being shown in Figures 3-5. One way to expand this would be to run the model I just mentioned. It would also be helpful in 6B-D to label what red or blue means (including labeled colors in B and C would help too). I know it's in the legend but it doesn't hurt to directly label to aid in making it quickly accessible.

– Page 16 line 10, what does developmental maturity mean? Please use gestation time or postnatal time or PMA to clarify what you mean here.

---

## [Author Response]

Essential revisions:Major Concerns:1. One important concern brought up by all three reviewers is the confusion with the terminologies GA, PMA, and post-natal time (PT), for preemies/full-term infants. Please define and simplify terms as suggested by the reviewers to ameliorate the readability and justify why the authors chose to use each of the terms at different times in the analyses. Overall, the reviewers suggested using GA and PT and replacing the analyses with GA to simplify the findings. Please see individual reviewers' comments to simplify the language and avoid confusion across the manuscript.

We are sorry for the confusion caused by the terminologies. We have replaced all the analyses involving PMA with GA, and thus only GA and PT were used in the manuscripts to simplify the terms and ameliorate the readability, except for Figure 1 which showed the general development trajectories of structural or functional properties in the visual cortex. Figures 2-6 and the corresponding results and discussions are all updated.

2. To make the results and the flow of the paper more cohesive, we recommend the authors frame and justify each analysis with a clear question that the analysis will answer or the hypothesis that the authors posit as it is often difficult to understand how an analysis being described would contribute to the overall claims of the paper.

Thanks for the suggestions, we have clarified descriptions of the hypothesis and questions for each of the analyses, and also added a summary of these questions and manuscript organization at the end of the introduction:

“In the present study, we first characterized the general development of structural morphology (CT) and microstructure (CM) in the ventral cortex and investigate the contributions of prenatal and postnatal time on the structural development. Then we used r-fMRI to characterize the innate organization of the ventral cortex as early as one day of life and investigated the effect of PT on functional networks of V1 and VOTC areas. Furthermore, we carried out a mediation analysis to investigate the relationship between functional and structural development in area V1. Finally, we evaluated the structural and functional differences of visual cortex between the term- and preterm-born babies, as the latter group had immature cortical development at birth but longer PT compared to the former group with equivalent PMA”.

The rationale for individual results was added before each of the analyses:

1) Before the section “General development of cortical thickness and cortical myelination in human infants” (figure 1), we added “To show the general developmental trajectories of CT and CM in the visual cortex during the human neonatal period” (Page 5 Line 19-20);

2) Before the section of “Contribution of postnatal time and gestational age on the development of cortical thickness and cortical myelination” (figure 2), we added “Similar developmental trajectories of CT and CM in the ventral cortex across PMA were observed in the above results. It would be interesting to know whether gestational age (GA, the time from fertilization to birth) and postnatal time (PT, the time from birth to MRI scan) would have different influences on their development because the PT and GA reflected two very factors (e.g. innate growth versus postnatal experience)” (Page 6 Line 19-21 and Page 7 Line 1-2).

3) Before the section of “Development of functional connections in the ventral visual cortex” (figure 4), we added

“Beyond structural development of the neonatal cortex, we further asked how the functional connectivity between the different visual sub-regions changes during early development” (Page 9 Line 11-13) and

“Based on the baseline functional connections in term-born neonates, we were intrigued to known how those functional properties in the ventral visual cortex develop during neonatal stage of human infants? If so, whether PT and GA had different influences on this process?” (Page 10 Line 11-14).

3. Relatedly, the reviewers also brought up important points in several places whereby the claims made in the paper are not supported by the analyses conducted under this claim (e.g.., in cortical myelination and overall functional connectivity analyses in Figure 2,4, and 5 as well as the final results). The statistical tests and analysis conducted need to support the conclusions and claims. We recommend that the authors rephrase or tone down the conclusions made in this analysis or reassess their analyses to support the claims. A similar concern is raised by reviewers 1 and 3 in the final section "Comparison between structural and functional properties…", as the reader is left expecting a formal comparison between structural and functional development. Please consider an analysis as suggested by reviewer 3 for e.g., a regression with functional connectivity and structural metrics, to ask for example if a region's homotopic connectivity is correlated with CT or CM and which aspect of time (gestation or postnatal) is most important.

We have added a formal analysis of the structural-functional relationship to examine if a region's homotopic connectivity is correlated with CT or CM and which aspect of time (gestation or postnatal). Please see details in response to R1Q1 below.

4. Given the authors used an adult atlas to draw regions on the infants' brains, a sanity check of how well the Glasser atlas regions align on a neonatal brain at e.g., 37 weeks and compare that to 42 weeks old and adult brain and running some statistics on how these cortical-based alignments from the adult brain to the neonatal brain align, for a few sample subjects is recommended.

We have newly added analysis using the Glasser’s atlas and compared the area V1 transformed from adult space and manually delineated V1 in the neonatal space. Please see the response to the R1Q5.

5. A similar sanity check is also essential to check the white-gray matter boundary segmentation in some sample infant data as the authors use cortical thickness as one of their measures and CT is measured as the distance between the pial surface and gray-white matter boundary and the cortical ribbon of infants is thin at birth, there may be a possibility that partial-volume effects could be more prevalent in less-developed infants and impact myelin metrics.

We have added a sanity check on the segmentation accuracy of white-gray boundary in Figure 1—figure supplement 3. See the response to R1Q5.

6. The reviewers also asked if the authors used Fisher Z to transform their correlations? This is important to clarify as a significant portion of the results in the paper are correlations and it is unclear from the methods whether the authors are transforming their correlation values to make that use appropriate.

We are sorry for the confusion. All the statistical analyses involving correlation coefficients were Fisher-Z transformed. We have clarified this point in the revised manuscript.

7. Since one of the big conclusions of this paper is that not all structural and functional aspects are affected by gestation or postnatal time the same way, having a quick summary of those findings would be helpful. The reviewers suggest a summary table showing the morphological and myelin measures against gestational time and prenatal time, whether the gestational or prenatal time was a significant modulator of that measure.

Thank you for this suggestion. A summary table (Table 1) has been added to the manuscript.

8. Reviewer 2 brings up an important point about the homotopy analysis that needs clarification and perhaps rethinking of the statistical tests (e.g., comparing 34 homotopic pairs with the hundreds of non-homotopic pairs) that must be addressed in the revision. Please also consider the recommended alternative test by reviewer 2 that would compensate for some of the constraints to measure the homotopic correlation for a region.

We have modified the homotopic analysis following the suggestions of reviewer 2, by defending bilateral connections in to three categories of homotopic connections, adjacent connections and distal connections. Please see the specific responses to R2Q9 below.

Reviewer #1 (Recommendations for the authors):While I find this work theoretically well-motivated and the use of the large dHCP dataset very exciting, there are some major concerns that must be addressed before this study is ready for publication:1. There was a lot of confusion for me about the terminology between GA, PMA, and post-natal time, for preemies/full-term infants. To simplify the matter, I would first define these ages for the reader and also choose to use a single term, preferably GA as postmenstrual age would be GA plus the chronological age. This would make it easy to follow the flow of the analysis, especially in the section: "Contribution of postnatal time on the development of cortical thickness and cortical myelination". I also feel the same about the analysis in figure 3, it would make it clearer to use GA and postnatal time, as PMA already included both and it's difficult to see the relationship of the GA on CT and CM separately.

We agree it will be clearer to use GA and postnatal time instead of PMA and have revised the manuscript throughoutly. Please see our response to the 1^st^ major concern in the Essential Revisions (for the authors) section above.

2. Relatedly, I really like Supplementary Figure 4c and 4f, and they may be your point on the postnatal age and GA influence on CT and CM really clear and they can be moved as one of the main figures in the paper.

Thanks for the Reviewer’s suggestion. We have updated the results relating to Supp Figure S4 following the suggestion of Reviewer 2 using a linear mixed-effect model instead of the present ANOVA analysis to keep the continuity of the data (Page 7 Line 15-19):

“We applied a linear mixed-effect model to test whether the CT (or CM) of the whole ventral cortex were differently influenced by the GA vs. PT, and found that the GA had a significantly stronger effect on the CM than PT (interaction between GA and PT, p < 0.05) but no significant difference between GA and PT effect was found on the development of CT (p > 0.6)”.

Therefore, the previous Supplementary Figure 4 no longer exists, but the related information is described in text.

3. It would be good to sanity check how the adult Glasser maps align on a neonatal brain at e.g., 37 weeks and compare that to 42 weeks old and adult and run some statistics on how these cortical-based alignments from the adult brain to the neonatal brain align, for a few sample subjects. Another sanity check that is necessary especially with the cortical thickness measure is to check the segmentations in the data. It would be useful to provide a montage of a few sample subjects showing the segmentation of the gray-white boundary as the contrast properties in infants is hard to decipher.

Thank you for the suggestions. We agree it is necessary to provide a sanity check of the registration accuracy.

Regarding the registration between adult Glasser’s maps and neonatal brain, we didn’t register the adult maps into individual spaces of the neonates since this would need multiple steps of registrations, resulting in inaccuracies and the additional burden of checking the registration of each subject. Instead, we first registered all neonatal surfaces into a neonate-specific template (40-week) using the transformation matrices provided by the dHCP pipeline based on the multimodal surface matching method (MSM, Makropoulos et al., 2018; Robinson et al., 2018). Then we performed a similar MSM registration to transform Glasser’s maps into the 40-week template. We visually checked the registration results for some of the typical sulci and gyri in Figure 1—figure supplement 1. Although the best way is to use a neonate-specific parcellation for the present analysis, there is no such fine cortical parcellation of human neonates yet. In addition, we used a definable area—the V1 area as an example to quantitatively evaluate the quality of the registration. Specifically, we manually delineated the area V1 on the gradient map (Figure 1—figure supplement 4a) of the averaged CM map across all neonates and calculate the dice coefficient between the transformed V1 from Glasser’s atlas and the manually delineated V1. The result showed a dice score of 0.83 and 0.80 for right and left hemispheres, respectively. As shown in Figure 1—figure supplement 4b, the location of the registered V1 and manually delineated V1 area was consistent, although the latter was slightly bigger. Identifying other visual areas in Glasser’s map would need task functional data that were not available in the present study. These quality checks have been added in the Methods section (Page 23 Line 23-29 and Page 24 Line 1-5).

Regarding the gray-white matter segmentation, we used the dHCP pipeline in this work. The segmentation procedure was included in the dHCP pipeline, and developers visually checked the quality of segmentation in their study, in which only 5% of the subjects had poor segmentation based on 160 random images from 37-44 weeks of PMA (Makropoulos et al., 2018). Here we provided some examples of the gray-white boundaries from the subjects scanned at different ages in Figure 1—figure supplement 3 (Page 22 Line 19).

4. In the first analysis are there any hemispheric differences in the relationship between CT and CM and age?

We applied a linear mixed-effect model to estimate the hemispheric difference in the relationship between structural measurements and age with the hemisphere, age, and the interaction between them as a fixed effect. The intercept and coefficient of the hemisphere were the random effects. For the averaged measurements across the whole ventral mask, the main effect of age was significant in both CT and CM (*p* < 0.05), but the main effect of the hemisphere and the interaction between age and hemisphere were not (*p* > 0.3).

5. Can the authors describe how the average cortical thickness and myelination maps in Figures1 b, c are generated? Are the individual baby brains aligned to a common surface? Or is it an average of each of the 34 rois, averaged across the babies and projected on a sample surface?

The average cortical thickness and myelination maps are generated by aligning individual baby brains to a common surface and then average. We have clarified this procedure in the Methods section (Page 26 Line 1-6):

“To generate the average cortical measurements (CT and CM) at the group level, we first registered the metrics of every subject onto a common 40-week template surface using the registration files provided by dHCP pipeline, and then averaged them in a vertex-wise manner across subjects within a specific week (e.g. 38-week map included neonates whose PMA ranged from 37.5 to 38.5 weeks) “.

6. I really liked the functional connectivity analysis, however, I am unclear what the non-homotopic means in this case. Can the authors also explicate what negative connectivity would mean and if it is significant what it would imply? It is also unclear how a correlation of r=.09 in Figure 4b would be significant and passes a multiple comparison threshold. Please clarify.

The non-homotopic indicated the pairs between two non-paired areas in two hemispheres, e.g., V1 in the left hemisphere and V2 in the right. In the present study, only 34 paired areas are homotopic (diagonal of the connectivity matrix) and all other connections were non-homotopic. Now we have changed them into three groups following the suggestion by Reviewer 2: “namely the homotopic connection (the connection between two paired areas in two hemispheres. e.g. right and left V1), adjacent connection (e.g., right V1 and left V2 as well as left V1 and right V2 since V1 and V2 are adjacent) and distant connections (two areas that were not the paired or adjacent).” We have added this description in the Methods section (Page 26 Line 24-29).

It’s always been controversial whether the negative functional connectivity (also called anti-correlation) is valid neurophysiologically or just analytic artifacts (Chai et al., 2012; Murphy et al., 2009). Some researchers might take the anti-correlation as an inhibitory interaction in the specific circuit if it was not driven by the analytic procedure (Keller et al., 2015; Meskaldji et al., 2016). In the revised paper, we excluded the negative connections to avoid such uncertainty.

Sorry for the mistake about the “*r* = 0.09 in Figure 4b”. We mistakenly calculated the p-value based on Pearson correlation using the number of time points as the degree of freedom. Now we updated the statistical analysis using an independent-sample T-test to test whether the homotopic correlation was significantly greater than zero across subjects, and the results showed that the homotopic connections in all ROIs of ventral cortex were significant (mean *r* = 0.13– 0.43, *t* > 12.87, *p* < 10^-9^; Figure 4a-b). See corrected parts in the Methods and Results (Page 9 Line 17-21).

7. In "Comparison between structural and functional properties in ventral visual cortex of term- and preterm-born infants" analysis: The juxtaposition of these two statements is very confusing to me: "For the structural measurements, 56 of 68 ROIs showed significant lower CT in the term-born than preterm-born infants (ts = -4.26 to -2.71; FDR q < 0.05; Figure 6b). Particularly, the term born infants showed significantly higher CT in the area V1 (t = -2.55, p < 0.02) but not VOTC (t = -1.24, p > 0.2; Figure 6f)." I would rather see a mean CT (mean/std) across the rois in numbers and how it varies between preterm and term-born infants.

Thanks for the suggestions. We added bar figures to show the mean CT and CM across all 68 ROIs bilaterally between term and preterm-born infants and revised the descriptions as “For the structural measurements, the mean CT across all ROIs was significantly lower in the term- (1.39 ± 0.05) than preterm-born infants (1.35 ± 0.05; *t* = -3.16, *p* < 0.01) and 10 of 68 ROIs showed significantly lower CT in the term-born than preterm-born infants (Figure 7—figure supplement 1). On the contrary, the mean CM across all ROIs was higher in the term-born (1.22 ± 0.10) than preterm-born (1.09 ± 0.08; *t* = 7.11, *p* < 10^-9^) infants and all ROIs were significantly higher in the term- than preterm-born groups.” in the Results section (Page 15 Line 3-9). Meanwhile, we found that the previous statement “56 of 68 ROIs showed significantly lower CT in the term-born than preterm-born infants” was problematic. The correct result is that 56 of 68 ROIs showed lower CT in the term-born than preterm-born infants and 10 of them were significant. We have corrected the statement in the above changes.

8. Relatedly in figure 6b-c, it is difficult to know the significant ts. Please change the non-significant ts to a different color or grays.

We have revised Figure 6b-c accordingly.

9. The analysis titled "Comparison between structural and functional properties in ventral visual cortex of term- and preterm-born infants" is a bit confusing to me. I am not clear on how we learn anything about the structural-functional relationship. It would be more convincing to see some comparative analysis between the structural and functional measures and how this varies by age, and in pre and full-term infants. The current analysis is not doing justice to the title of this analysis.

Please see the response to R1Q1.

10. Your results of more myelination in the V1 than higher-order areas are consistent with the hierarchical findings of a prior paper on infants in their first six months of life (see Natu, et al., 2021, Nature Communications Biology). What is interesting to see is that neural scaffolding starts to develop early on during prenatal life. However, it is also interesting that you found that the cortical myelination in the ventral cortex was not directly influenced by postnatal time and I think these points can be discussed in the discussion.

Thanks for the advice, we added the following paragraph in the Discussion section (Page 18 Line 17-23):

“However, we found a significant correlation between CM in the ventral cortex and GA but not PT in the neonatal period, suggesting that early cortical myelination was primarily determined by the endogenic growth in the uterine environment. The hierarchical pattern of CM along the ventral visual cortex agreed with a previous study on infants at around six months of age (Natu et al., 2021). Our results further suggested that this hierarchical pattern occurs as early as the neonatal period.”

11. Finally, I am not entirely clear if the term postnatal "experience" used throughout the manuscript, is appropriate in this case, perhaps the authors can tone it down the postnatal time, as we don't have any behavioral measure of visual experience per se, like contrast properties, acuity, etc.

We have toned it down to the postnatal time or postnatal environment in the revised article.

Reviewer #2 (Recommendations for the authors):Most of my recommendations are integrated into the public comments. That said, I wish to emphasize how important I think this question is and the impressive amount of work the authors have already done.A general style comment is that I had trouble following the thread in the results. I think that each analysis should be framed more clearly in terms of the question it is answering. It was often difficult to understand how an analysis being described would contribute to the overall claims of the paper and what alternative hypotheses were possible for the analysis. This led to what felt like a list of results, rather than a coherent narrative.

Thanks for the suggestions, we have added clearer descriptions of the hypothesis and questions for each of the analyses to put them in a coherent story in the revised manuscript. Please see the Response to the 2^nd^ major concern in the Essential Revisions (for the authors) section above.

Recommended analyses:Regarding point 3: An alternative to this analysis I think the authors ought to consider is to control PT (i.e., find preterm and term groups that have an equivalent postnatal age) and see how variability in GA contributes to the metrics they observe. This reverses the logic of the analysis the authors did, but does so in a way I think is tightly controlled.

We agree with this view, but the present data might not be suitable for this analysis. The range of the PT in the term-born infants was from 0 to 6.72 weeks (mean value: 1.12 ± 1.25) but PT in the preterm-born infants was from 0.28 to 19.72 weeks (mean value: 8.91± 4.42). As shown in Author response image 1, the distribution of PT was also very different in the term and preterm groups. Therefore, we simply controlled for GA or PT in the term-born infants to see the effect of the other one (Page 7 Line 20-28, and Figure 2 e-h).

**Author response image 1. sa2fig1:** The distribution of neonates with different PT in the term and preterm infants.

Regarding point 9: an alternative test that compensates for some of these constraints is to get the homotopic correlation for a region and then subtract it from the average of the non-homotopic correlations for that region (could even just use the distal pairs for this). This difference can then be compared across individuals for each region.

We have performed a non-parameter statistical analysis (Wilcoxon signed-rank) to statistically compare homotopic connections and other types of connections. Please see the Response to R2Q9.

Regarding point 10: the authors should consider using the Wang et al., (2015) atlas, since that more carefully divides visual cortex.

The difference between Glasser’s atlas and Wang’s atlas is that Wang’s atlas is focused on the human visual cortex and provided 25 areas per hemisphere, including 8 early visual cortex (e.g. V1-3), 4 lateral occipital areas (e.g. LO1, LO2), 5 ventral visual cortex (e.g. hV4, PHC), 7 parietal regions (e.g. IPS) and 1 fontal area (e.g. hFEF). While the Glasser’s atlas contained 180 areas per hemisphere for the whole brain cortex with 17 ROIs in the ventral visual cortex p. Wang’s atlas provided a more detailed parcellation than Glasser’s atlas in the early visual cortex (e.g. the area V1 was divided into V1v and V1d in the Wang’s atlas). Therefore, we tried to use Wang et al., (2015) atlas to reanalysis the data.

We registered Wang’s atlas onto 40-week neonatal template by the same method applied to Glasser’s atlas (see Author response image 2), and then calculated the homotopic connection for the 17 ROIs within the visual cortex in the Wang’s atlas (we excluded areas in the parietal and frontal lobe). The resulting connections ranged from 0.001 to 0.55, and the mean homotopic connection was 0.23 (SD = 0.17). We found that the correlation between bilateral V1 was improved while some small ROIs showed very low correlations. Those low value might be due to the registration inaccuracy that these refined areas were misregistered into non-homologous positions between hemispheres, and therefore, we think Wang’s atlas with very small ROIs may not be appropriate for the task here.

**Author response image 2. sa2fig2:** Mapping of Wang et al. (2015) atlas on the 40-week neonatal template.

Reviewer #3 (Recommendations for the authors):– The language at times gets confusing referring to things such as PMA, GA, PA, etc. So I would simplify and clarify the language and say "gestational time" or "gestational age" when referring to how long a neonate spent gestating and "postnatal time" when referring to how much experience it has had after birth. Gestational time and postnatal time are not that long character-wise and will go a long way in improving the readability of the manuscript. I found myself having to go back and look up acronym meanings several times.

We have defined GA as the "gestational age" referring to the period a neonate spent gestating and PT as the "postnatal time" referring to how much experience it has had after birth. Since we have throughoutly revised the analyses using only GA and PT as the developmental factors, we think it would make the manuscript easier to read now.

– In figure 2, the outline is hard to see in some panels. For example, in Figure 2a, it at first looks like only that central fusiform ROI is the only ROI with a significant value, but I didn't realize until later that it's actually all of the other ROIs that seem to be outlined and the fusiform ROI is not significant. I would say that if an ROI is not significant, I would just black it out, that way if an ROI is colored it is because that value is significant. This will be more straightforward than the current outlines.

We have updated figure 2 in the new manuscript to only show the significant ROIs, and also incorporated comments from all reviewers.

– Supp Figure 4 seems important why isn't it a main figure or part of main figure 3? 3 could be condensed or pruned to make room for it. Also on the topic of supplementary figure 4, from what ROI was this taken? All of the visual HCP-MMP ROIs (like V1 + high-level regions) or was it one particular region? If incorporated into Figure 3, showing the data for cortical myelination like parts D and E would be great.

Thanks for the Reviewer’s suggestion. We have updated the results relating to Supp Figure S4 following the suggestion of Reviewer 2 using a linear mixed-effect model rather than the present ANOVA analysis to keep the continuity of the data (Page 7 Line 15-19):

“We applied a linear mixed-effect model to test whether the CT (or CM) of the whole ventral cortex were differently influenced by the GA vs. PT, and found that the GA had a significantly stronger effect on the CM than PT (interaction between GA and PT, p < 0.05) but no significant difference between GA and PT effect was found on the development of CT (p > 0.6).”

Therefore, the previous Supplementary Figure 4 no longer exists.

– Figure 4d, would it be worthwhile to make this figure comparing infants with longer PMA times since it was shown that functional connectivity network properties change with development? It might be nice to see how and if the network in Figure 4D changes (do some regions get closer together, etc).

We add a new analysis to directly evaluate the developmental change of the functional connections in the visual cortex with PMA (Page 11 Line 15-19):

“The correlations between ipsilateral connections and PMA showed both positive and negative results in different regions of the ventral cortex. Particularly, the early visual cortex (e.g. V1 and V2) showed decreasing connection to other visual areas across PMA while part areas in the VOTC (e.g. fusiform and parahippocampus) showed increasing tendency (Figure 4—figure supplement 4).”

– Typo in figure titles in Figure 5.

We have revised accordingly.

– Page 15 Line 3 states that term-born infants have a lower cortical thickness, but then on line 5 it says that term-born infants have a higher cortical thickness in V1. I think this is a typo, Figure 6F would suggest term-born have thinner V1.

We have corrected the typo. The term-born infants had a lower cortical thickness in V1 than preterm-born infants.

– On the topic of Figure 6, it is being shown that the cortex is thinner in V1 in term-born infants who have comparatively less postnatal time. It is also shown that these term-born infants have more myelin. Typically, myelin increases with development, and the cortical thinning that occurs later in childhood is thought to be a result of increasing myelination at the gray-white matter border (see Natu et al. 2019 PNAS). Is it possible some of this myelin difference could result from the fact that in term-born infants the cortex is thinner (1.4mm in V1) and thus some voxels could partial-volume white matter voxels more easily? I know that white matter in neonates appears darker than the cortex in a T1-weighted image, but just want to bring up this potential point that a thinner cortex, given constant-sized voxels, is more likely to be biased by white matter signal from voxels sampling deep cortical layers.

We agree that there is a possibility that thinning of the cortex and change of the cortical contrast due to myelination could contribute to our observation. Although we have done our best to ensure the segmentation accuracy (see the response to R1Q5), there’s no way to exclude this effect. We added a paragraph in the limitation section to describe the possible bias driven by the thinner cortex in the term-born infants compared to preterm-born infants (Page 20 Line 20-24):

“And also the term-born infants, who had thinner cortex compared to preterm-born infants, were more likely to be affected by the partial-volume effect, which may contribute to the myelination difference observed between groups.”

– It might be nice to have a summary table showing, for each measure you compared against gestational time and prenatal time, whether the gestational or prenatal time was a significant modulator of that measure. Since one of the big conclusions of this paper is that not all structural and functional aspects are affected by gestation or postnatal time the same way, having a quick summary of those findings would be helpful.

We appreciate the reviewer’s suggestion, and we have provided a summary table.

– In the last Results section "Comparison between structural and functional properties…", I was expecting the authors to more formally compare structural and functional development. I was expecting them to do a regression with functional connectivity and structural metrics, to ask for example if a region's homotopic connectivity is correlated with CT or CM and which aspect of time (gestation or postnatal) is most important. You could even model gestational time and postnatal time separately. I say this because the data currently in Figure 6 while certainly useful, are summaries of data already being shown in Figures 3-5. One way to expand this would be to run the model I just mentioned. It would also be helpful in 6B-D to label what red or blue means (including labeled colors in B and C would help too). I know it's in the legend but it doesn't hurt to directly label to aid in making it quickly accessible.

Following the suggestions, we have added a formal analysis to investigate the relationship between structural and functional development in area V1. Please see the Response to the 1^st^ concern of Reviewer 1 (public review) for more details.

– Page 16 line 10, what does developmental maturity mean? Please use gestation time or postnatal time or PMA to clarify what you mean here.

We replaced developmental maturity with GA in the manuscript to make it clear.

References:

Chai XJ, Castañán AN, Öngür D, Whitfield-Gabrieli S. 2012. Anticorrelations in resting state networks without global signal regression. Neuroimage 59:1420–1428. doi:10.1016/j.neuroimage.2011.08.048

Fjell AM, Grydeland H, Krogsrud SK, Amlien I, Rohani DA, Ferschmann L, Storsve AB, Tamnes CK, Sala-Llonch R, Due-Tønnessen P, Bjørnerud A, Sølsnes AE, Håberg AK, Skranes J, Bartsch H, Chen CH, Thompson WK, Panizzon MS, Kremen WS, Dale AM, Walhovd KB. 2015. Development and aging of cortical thickness correspond to genetic organization patterns. Proc Natl Acad Sci U S A 112:15462–15467. doi:10.1073/pnas.1508831112

Kamps FS, Hendrix CL, Brennan PA, Dilks DD. 2020. Connectivity at the origins of domain specificity in the cortical face and place networks. Proc Natl Acad Sci U S A 117:6163–6169. doi:10.1073/pnas.1911359117

Keller JB, Hedden T, Thompson TW, Anteraper SA, Gabrieli JDE, Whitfield-Gabrieli S. 2015. Resting-state anticorrelations between medial and lateral prefrontal cortex: Association with working memory, aging, and individual differences. Cortex 64:271–280. doi:10.1016/j.cortex.2014.12.001

Makropoulos A, Robinson EC, Schuh A, Wright R, Fitzgibbon S, Bozek J, Counsell SJ, Steinweg J, Vecchiato K, Passerat-Palmbach J, Lenz G, Mortari F, Tenev T, Duff EP, Bastiani M, Cordero-Grande L, Hughes E, Tusor N, Tournier JD, Hutter J, Price AN, Teixeira RPAG, Murgasova M, Victor S, Kelly C, Rutherford MA, Smith SM, Edwards AD, Hajnal J V., Jenkinson M, Rueckert D. 2018. The developing human connectome project: A minimal processing pipeline for neonatal cortical surface reconstruction. Neuroimage 173:88–112. doi:10.1016/j.neuroimage.2018.01.054

Meskaldji DE, Preti MG, Bolton TA, Montandon ML, Rodriguez C, Morgenthaler S, Giannakopoulos P, Haller S, Van De Ville D. 2016. Prediction of long-term memory scores in MCI based on resting-state fMRI. NeuroImage Clin 12:785–795. doi:10.1016/j.nicl.2016.10.004

Murphy K, Birn RM, Handwerker DA, Jones TB, Bandettini PA. 2009. The impact of global signal regression on resting state correlations: Are anti-correlated networks introduced? Neuroimage 44:893–905. doi:10.1016/j.neuroimage.2008.09.036

Natu VS, Rosenke M, Wu H, Querdasi FR, Kular H, Lopez-Alvarez N, Grotheer M, Berman S, Mezer AA, Grill-Spector K. 2021. Infants’ cortex undergoes microstructural growth coupled with myelination during development. Commun Biol 4:1–12. doi:10.1038/s42003-021-02706-w

Robinson EC, Garcia K, Glasser MF, Chen Z, Coalson TS, Makropoulos A, Bozek J, Wright R, Schuh A, Webster M, Hutter J, Price A, Cordero Grande L, Hughes E, Tusor N, Bayly P V., Van Essen DC, Smith SM, Edwards AD, Hajnal J, Jenkinson M, Glocker B, Rueckert D. 2018. Multimodal surface matching with higher-order smoothness constraints. Neuroimage 167:453–465. doi:10.1016/j.neuroimage.2017.10.037

Wang F, Lian C, Wu Z, Zhang H, Li T, Meng Y, Wang L, Lin W, Shen D, Li G. 2019. Developmental topography of cortical thickness during infancy. Proc Natl Acad Sci U S A 116:15855–15860. doi:10.1073/pnas.1821523116